# Enhanced mitochondrial biogenesis promotes neuroprotection in human pluripotent stem cell derived retinal ganglion cells

Michelle Surma[1,6], Kavitha Anbarasu[1,2,6], Sayanta Dutta[1], Leonardo J. Olivera Perez[3], Kang-Chieh Huang[4], Jason S. Meyer[1,2,3,5] & Arupratan Das [1,2,5✉]

Mitochondrial dysfunctions are widely afflicted in central nervous system (CNS) disorders with minimal understanding on how to improve mitochondrial homeostasis to promote neuroprotection. Here we have used human stem cell differentiated retinal ganglion cells (hRGCs) of the CNS, which are highly sensitive towards mitochondrial dysfunctions due to their unique structure and function, to identify mechanisms for improving mitochondrial quality control (MQC). We show that hRGCs are efficient in maintaining mitochondrial homeostasis through rapid degradation and biogenesis of mitochondria under acute damage. Using a glaucomatous Optineurin mutant (E50K) stem cell line, we show that at basal level mutant hRGCs possess less mitochondrial mass and suffer mitochondrial swelling due to excess ATP production load. Activation of mitochondrial biogenesis through pharmacological inhibition of the Tank binding kinase 1 (TBK1) restores energy homeostasis, mitigates mitochondrial swelling with neuroprotection against acute mitochondrial damage for glaucomatous E50K hRGCs, revealing a novel neuroprotection mechanism.

[1] Department of Ophthalmology, Eugene and Marilyn Glick Eye Institute, Indiana University, Indianapolis, IN 46202, USA. [2] Department of Medical and Molecular Genetics, Indiana University, Indianapolis, IN 46202, USA. [3] Indiana University School of Medicine, Indianapolis, IN 46202, USA. [4] Department of Biology, Indiana University Purdue University, Indianapolis, IN 46202, USA. [5] Stark Neurosciences Research Institute, Indiana University, Indianapolis, IN 46202, USA. [6] These authors contributed equally: Michelle Surma, Kavitha Anbarasu. ✉email: arupdas@iu.edu

Neurons require an increased supply of ATP compared to other cells in order to establish membrane potential for neurotransmission and synaptic activity[1]. Oxidative phosphorylation (OXPHOS) in mitochondria is the primary source for cellular ATP, thus mitochondrial dysfunction often associates with the neurodegenerative diseases such as mitochondrial optic neuropathies[2], Parkinson's disease, and amyotrophic lateral sclerosis[4]. Among CNS neurons, retinal ganglion cells (RGCs) of the optic nerve require a steady source of ATP for their varying action potential firing frequency requirements and due to the existence of long partially unmyelinated axons[5]. Axons of human RGCs remain unmyelinated after leaving the cell body in the retinal layer as they travel through the optic nerve head, until they reach the posterior end of lamina cribrosa[6]. Action potentials through the unmyelinated axons propagate slowly and require more ATP[7]. Thus, human RGCs are the most sensitive towards mitochondrial dysfunctions. As such, inherited mutations in the complex I subunit of electron transport chain cause Leber's hereditary optic neuropathy[8], OPA1 mutations of the mitochondrial inner membrane protein required for mitochondrial fusion cause dominant optic atrophy[9]; together these diseases are known as mitochondrial optic neuropathies.

Increased intraocular pressure (IOP) has been the major risk factor for glaucoma, however there are glaucoma patients with normal IOP, often referred to as normal-tension glaucoma (NTG)[10]. NTG patients are particularly challenging to treat as sufficient lowering of IOP is not always possible and a RGC protection strategy is yet to be discovered. Mutations in the mitochondrial homeostasis pathways are highly prevalent among NTG patients which includes the dominant Optineurin (OPTN$^{E50K}$) mutation. OPTN is a critical player for mitophagy and the E50K mutation is the most prevalent (~17%) among NTG patients[11,12]. Intriguingly, these inherited mitochondrial mutations, though present in every cell of a patient's body, only cause death to RGCs, leading to vision loss. Thus, RGCs serve as a great neuronal model for investigating mechanisms of improving mitochondrial homeostasis for developing neuroprotection, which may be applicable to other CNS neurons. MQC involves biogenesis of healthy mitochondria, fusion, fission, and degradation of damaged mitochondria (mitophagy). Thus, reduced mitochondrial homeostasis is expected in the E50K RGCs. However, to date we have a very limited understanding on how to improve MQC to enhance RGC viability. RGCs spontaneously degenerate at a slow rate during aging, however this degeneration is exacerbated during optic neuropathies[6]. Thus, identification of mechanisms that will improve RGC survival could provide therapy for optic neuropathies as well as for age-related ganglion cell loss.

AMP-activated protein kinase (AMPK) is the master energy sensor in cells[13]. Declines in energy homeostasis are associated with aging[14], obesity[15] and neurodegenerative conditions[16]. Thus, it is expected that a decline in mitochondrial homeostasis could lead to energy stress in these disease conditions. Energy stress activates AMPK which activates peroxisome proliferator-activated receptor gamma co-activator 1α (PGC1α)[17], which is the master regulator of mitochondrial biogenesis. Under normal conditions, TBK1 phosphorylates the inhibitory site of AMPK[18] limiting energy expenditure, which becomes counterproductive during aging and obesity. Conversely, TBK1 inhibition has been found to increase energy expenditure, reduce obesity and insulin resistance in mice[15]. It remains unexplored if targeting TBK1 could activate biogenesis of healthy mitochondria leading to restoration of energy homeostasis and enhanced RGC survival under disease conditions.

Here, we used a robust and well characterized human stem cell differentiated RGC (hRGC) model to investigate if improving energy homeostasis through the TBK1-AMPK pathway could promote MQC and neuronal survival. It is critical to identify mechanisms for improving MQC in human RGCs as mice RGCs are very different from human both anatomically and by genetic makeup[19]. Differentiated hRGCs used here are well characterized and show the hallmark of human RGCs, as characterized by their morphological features, gene-expression pattern, and electrophysiological properties such as spontaneous generation of action potentials[20,21] and ability to graft in the mice retina[22]. We have shown here that under acute mitochondrial damage hRGCs activate the mitochondrial biogenesis pathway to maintain mitochondrial homeostasis. Furthermore, we harnessed the biogenesis activation pathway by pharmacological inhibition of TBK1 mediated activation of AMPK-PGC1α biogenesis axis in hRGCs. Remarkably, activation of mitochondrial biogenesis reduced metabolic burden in the glaucomatous E50K hRGCs leading to recovery of mitochondrial swelling and pro-survival responses as an indication for neuroprotection.

## Results

### RGCs are highly efficient at degrading damaged mitochondria and producing healthy mitochondria to maintain homeostasis under stress.

To investigate mitochondrial homeostasis mechanisms in human RGCs we have used human stem cell differentiated RGCs (hRGCs) following a robust method[20,23] where cells express neuronal marker Tubulin β3 and RGC specific marker RBPMS (Supplementary Fig. 1a, b). Here, we have developed a "disease in a dish" model for glaucoma using CRISPR-engineered human embryonic stem cells with glaucomatous OPTN$^{E50K}$ mutation (H7-ESCs-E50K) (Supplementary Fig. 1c), patient-derived induced pluripotent stem cells with the E50K mutation (iPSCs-E50K)[24] and corresponding H7-ESCs and iPSC-E50K corrected (iPSC-E50Kcorr)[25] reporter lines for isogenic controls. As RGCs rely on a high ATP supply, it is likely that RGCs will be vulnerable towards mitochondrial damage since they are the primary source for cellular ATP. We asked if hRGCs can maintain mitochondrial homeostasis under acute damage. Carbonyl cyanide m-chlorophenyl hydrazone (CCCP) has been widely used in literature at 5–20 μM concentrations to induce mitochondrial damage[26,27]. This has been shown, 10 μM of CCCP is effective in changing mitochondrial morphology within a few seconds in cultured cells[27]. We have previously performed a dose response and shown 10 μM CCCP as minimum dose to have maximum mitochondrial degradation in hRGCs[23]. Here, we used 10 μM CCCP to cause acute mitochondrial damage for investigating how hRGCs restore mitochondrial homeostasis under stress. We tested whether hRGCs maintain the same total mitochondrial mass during CCCP damage by immunofluorescence against the mitochondria-specific protein Tom20 and by direct measurement of mitochondrial DNA copy number. Interestingly, we found that the total mitochondrial mass fluctuates around DMSO control condition during the course of CCCP treatment for both the WT and E50K hRGCs (Fig. 1a–d). We further validated this observation by qPCR-based measurement of mitochondrial DNA copy number relative to nuclear DNA and found WT hRGCs maintained mitochondrial mass similar to the control condition while glaucomatous E50K hRGCs showed moderate reduction at the longer CCCP treatment (Fig. 1e, f). These data suggest hRGCs possess stringent MQC mechanisms for maintaining total mitochondrial mass even with acute damage. However, in these experiments we measured total mitochondrial mass which cannot distinguish between the CCCP damaged degrading mitochondria versus the newly synthesized population.

While hRGCs maintained total mitochondrial mass under acute CCCP damage, they potentially have positive feedback

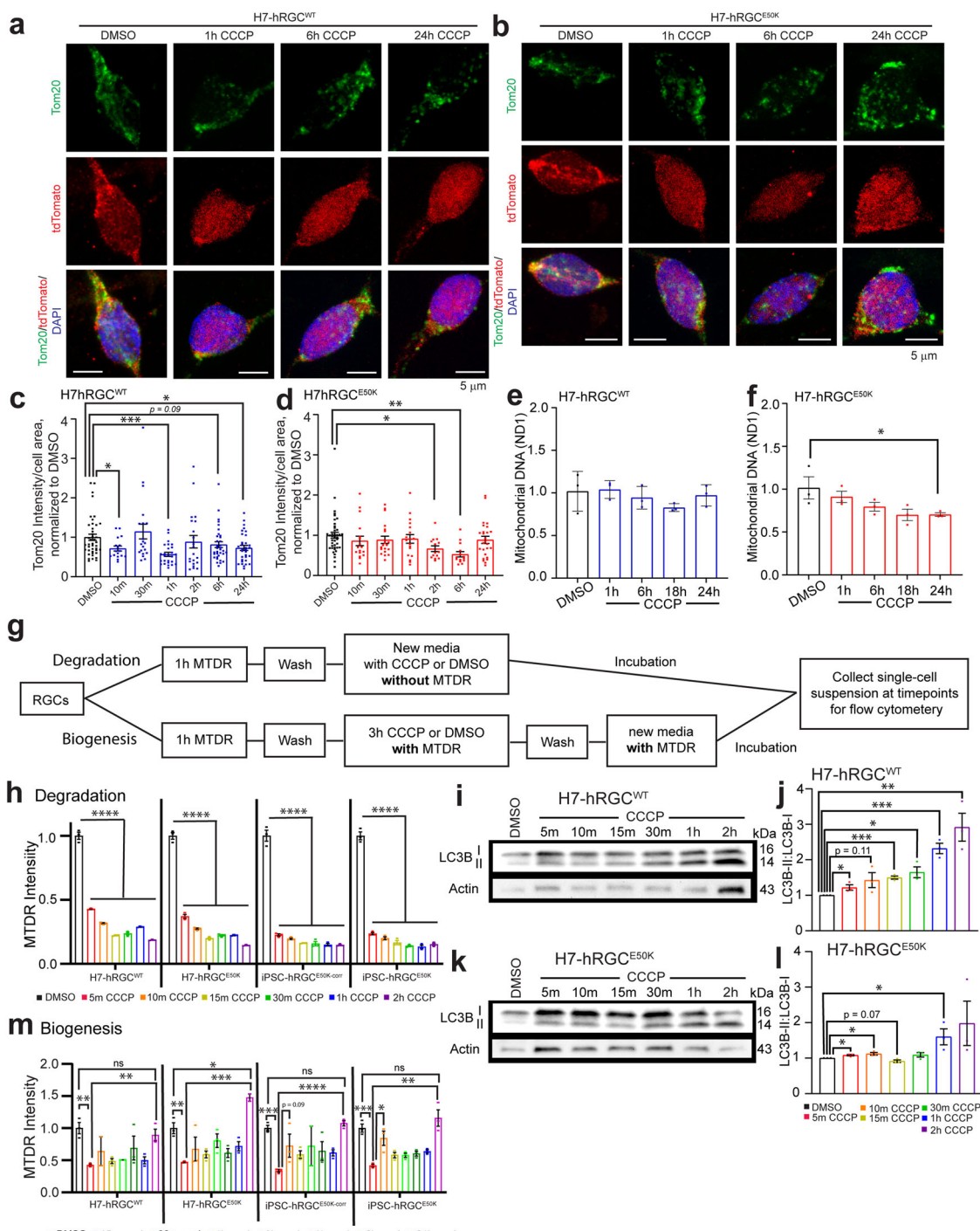

**Fig. 1 hRGCs maintain mitochondrial mass by simultaneous degradation and biogenesis of mitochondria under damage. a**, **b** Representative confocal immunofluorescence images of H7-hRGCs against mitochondrial Tom20, tdTomato, and DAPI after indicated 10 µM CCCP treatment timepoints. Scale bars are 5 µm. **c**, **d** Quantification of Tom20 intensity per cell area from sum projections of confocal z-stacks, normalized to corresponding DMSO. $n = 14$–42 cells per condition. **e**, **f** Mitochondrial DNA copy number analyzed by qPCR for mitochondrial ND1 gene relative to nuclear RNase P. Results shown as ΔΔCt fold changes relative to DMSO control for different times points of 10 µM CCCP treatment. $n = 3$, 3 technical repeats averaged for each biological repeat. **g** Schematic of MTDR flow experiments for tracking degrading versus newly synthesized mitochondria. **h** Mitochondrial mass in single cells at different CCCP (10 µM) treatment time points for degradation or **m** different timepoints post CCCP (10 µM) wash for biogenesis were measured by flow cytometer, and the average fluorescence intensity normalized to DMSO was plotted. $n = 3$. **i**, **k** Representative western blot images of LC3B and actin from **i** hRGC[WT] and **k** hRGC[E50K] treated with CCCP (10 µM) for the indicated timepoints. **j**, **l** Quantification of the ratio of LC3B-II to corresponding LC3B-I for each condition. $n = 3$. *$p$-value < 0.05, **$p$-value < 0.01, ***$p$-value < 0.001, ****$p$-value < 0.0001. Unpaired student's *t-test* between independent datasets. Error bars are SEM.

where cells degrade damaged mitochondria, which activates the biogenesis pathway to compensate the loss in order to maintain energy homeostasis. Measuring total mitochondrial mass in a cell will not resolve the newly synthesized versus the degrading population. To directly measure mitochondrial degradation and biogenesis, we used a mitochondria specific live cell dye MitoTracker Deep Red (MTDR) followed by flow cytometry-based measurements, a protocol previously developed by us[23] but with modifications as explained in Fig. 1g. The flow cytometry gating strategy is explained in Supplementary Fig. 2. MTDR passively diffuses inside the mitochondria and is retained based on active membrane potential, then the dye covalently binds to the thiol groups of cysteine residue containing proteins[28]. Hence, once labeled, any changes in the MTDR intensity should reflect the change in mitochondrial mass independent of mitochondrial membrane polarity. For measuring CCCP-induced mitochondrial degradation, we first labeled existing mitochondria in hRGCs with MTDR followed by a wash step and CCCP addition, and then flow cytometry-based measurements of mitochondrial mass in single cells at different time points (Fig. 1g). In this study design, we measured MTDR labeled mitochondrial mass in independent samples after different CCCP treatment timepoints. This allowed us to track degradation of the labeled population while excluding any newly synthesized mitochondria, as there was no dye in the media for new mitochondria to incorporate. Quantification of MTDR intensity among the WT and *E50K* hRGCs showed a prompt (within 5 min) degradation of mitochondria upon CCCP treatment (Fig. 1h). Damaged mitochondria are degraded by lysosomes via LC3B autophagic flux, a process known as mitophagy[29]. The hallmark for activation of the mitophagy pathway is an increase in lipidated LC3B. Mitophagy complex formation on mitochondria depends on the LC3B-I (non-lipidated, 16 kDa) to LC3B-II conversion (lipidated, 14 kDa), which migrates faster during gel electrophoresis and enrichment of the second band in western blot is a classic measurement of the induction of mitophagy[30]. We hypothesized that induction of mitochondrial damage would lead to mitophagy by increasing LC3B lipidation. Indeed, we observed rapid induction of mitophagy within 5 min of CCCP treatment as shown by increased lipidated LC3B form in both WT (Fig. 1i, j) and *E50K* hRGCs (Fig. 1k, l). Of note, we had previously shown lysosome-dependent mitochondrial degradation in hRGCs is essential for its viability[23]. It has also been shown that 10 μM of CCCP treatment for 5 min induces mitochondrial fragmentation for mitophagy in cultured HeLa cells and in mouse embryonic fibroblasts further supporting our observations[27]. Since total mitochondrial mass fluctuated close to the control condition under CCCP damage and the above data identified mitochondrial degradation under CCCP treatment by mitophagy, we asked if this mitochondrial loss was compensated for by the biogenesis of new mitochondria.

For measuring mitochondrial biogenesis, we have modified the flow cytometry experimental design to detect newly synthesized mitochondria. In this experimental design, we first labeled mitochondria with MTDR, then treated cells with CCCP to cause mitochondrial damage followed by a wash and media exchange containing MTDR but no CCCP, and then took mitochondrial mass measurements at different timepoints (Fig. 1g). Here, after CCCP wash, the newly synthesized mitochondria will incorporate MTDR, and we expect to see a gradual increase in MTDR labeled mitochondrial mass over time. We found compared to DMSO control, 15 min after the CCCP wash there is a significant loss of labeled mitochondrial population, indicative of degradation (Fig. 1m). Please note, mitochondria synthesized during the CCCP treatment are unlikely to be detected by MTDR as CCCP depolarizes

mitochondrial membrane potential and MTDR retention requires intact membrane potential. However, after CCCP wash MTDR should bind to the newly synthesized mitochondria, and indeed we observed hRGCs gradually increased mitochondrial mass post CCCP wash and reached levels similar to the control condition after 24 h (Fig. 1m). These data suggest hRGCs possess highly efficient MQC mechanisms where they efficiently degrade damaged mitochondria and produce healthy mitochondria to compensate the loss, presumably maintaining their energy homeostasis.

To visualize the biogenesis and degradation events, we monitored mitochondrial turnover using live cell JC1 dye which fluoresces green (monomers) when bound to damaged (depolarized) and red (J aggregates) when bound to healthy (polarized) mitochondria[31]. We expect, under biogenesis activation hRGCs will appear with more JC1 red mitochondria, and under mitochondrial damage cells will have more JC1 green mitochondria. To visualize polarized versus depolarized mitochondrial distribution at the basal level, we labeled mitochondria with the JC1 dye and acquired live cell images, then added CCCP in the presence of the dye, to monitor change in respective populations (Fig. 2a). We observed at the basal level (before CCCP) significant amounts of both JC1 labeled red and green mitochondria (Fig. 2b) indicating a polarized and depolarized mitochondrial distribution. This observation is supported by cultured healthy brain neurons that showed mitochondrial distribution with significant amounts of depolarized (JC1 green) and polarized (JC1 red) membranes[32]. Interestingly, we observed an increase in JC1 red intensity, but a more fragmented network and decrease in JC1 green mitochondria for both the WT and *E50K* hRGCs upon CCCP damage (Fig. 2b) leading to increased red-to-green ratio (Fig. 2c). This data is counterintuitive as we expected CCCP damage will increase JC1 green mitochondria with a reduction of healthy mitochondria (red). However, this observation could be supported if hRGCs efficiently clear CCCP-damaged mitochondria and simultaneously produce more to compensate the loss. Conversely, if we wash CCCP and allow cells to recover, hRGCs should restore their depolarized to polarized mitochondrial distribution like the basal level. To test this, we have performed a recovery experiment where we first treated both WT and *E50K* hRGCs with CCCP or DMSO (control). Next, we washed CCCP and incubated with JC1 media for recovery, then exchanged media without JC1 dye and acquired live cell imaging to measure polarized to depolarized mitochondrial distribution (Fig. 2d). Indeed, we observed WT hRGCs restored the mitochondrial distribution post CCCP wash similar to the control (DMSO) condition (Fig. 2e, f). Although, at the basal level WT and *E50K* hRGCs possess similar healthy to damaged mitochondrial ratio, after recovery from the CCCP damage, *E50K* hRGCs showed reduced healthy mitochondrial mass compared to its control condition (Fig. 2e, f). This suggests that WT hRGCs possess a distribution of healthy and damaged mitochondria at the basal level and are highly efficient in clearing acutely damaged mitochondria while producing new to compensate the loss. This balance between the degradation and biogenesis pathways helps hRGCs to regain mitochondrial homeostasis, however glaucomatous *E50K* hRGCs struggle to restore homeostasis after acute damage.

**hRGCs activate the biogenesis signaling pathway under mitochondrial damage.** Mitochondrial biogenesis has been shown to induce muscle growth after exercise[33] and synaptic plasticity in neurons[34]. However, it is unknown if mitochondrial damage triggers activation of the mitobiogenesis pathway in CNS neurons. AMP-activated protein kinase (AMPK) is a key regulator of energy

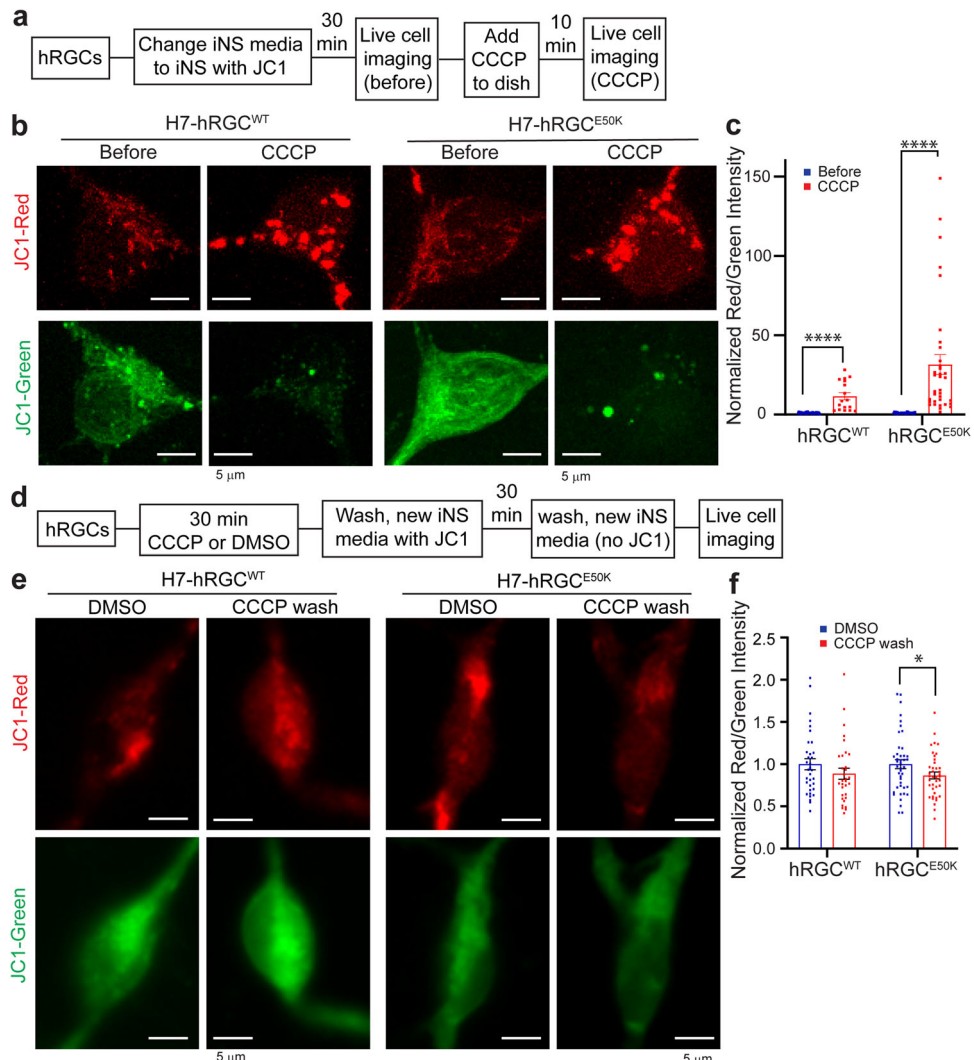

**Fig. 2 Under acute damage hRGCs possess efficient mitochondrial turnover mechanisms. a** Schematic of JC1 experiment to monitor mitochondrial degradation and biogenesis under CCCP stress. **b** Representative confocal images of JC1 labeled mitochondria in hRGCs before and after 10 min CCCP (10 μM) treatment. JC1 dye was added 30 min before CCCP and was present during treatments. Scale bars are 5 μm. **c** Quantification of the red-to-green fluorescence intensity ratios from the sum projections of confocal z-stacks, normalized to the corresponding before treatment values. $n = 17$-42 hRGCs. **d** Schematic of JC1 experimental design to study restoration of mitochondrial homeostasis after CCCP washout. **e** Representative confocal images of JC1 labeled mitochondria in hRGCs treated with DMSO or CCCP (10 μM) for 30 min, washed, labeled with JC1 dye, washed again, and then imaged without JC1 dye. **f** Quantification of the red-to-green fluorescence intensity ratios from sum projections, normalized to respective DMSO control. $n = 32$-43 hRGCs. *$p$-value < 0.05, **$p$-value < 0.01, ****$p$-value < 0.0001. Unpaired student's $t$-test between independent datasets. Error bars are SEM.

sensing and responds by increasing mitochondrial biogenesis[35]. AMPK is a heterotrimeric protein activated by the phosphorylation of Thr172 in the AMPKα subunit[36] which in turn activates PGC1α[15] at Ser571. PGC1α is the master regulator of mitochondrial biogenesis[37] and is a transcriptional co-activator of nuclear respiratory factor 2 (NRF2) and NRF1 which activate transcription factor A, mitochondrial (TFAM) for mitochondrial DNA (mtDNA) gene expressions[38]. The biogenesis pathway can also be activated by the transcriptional co-activators PGC1β and PGC1-related co-activator (PRC)[39]. We asked if mitochondrial damage leads to activation of the AMPKα–PGC1α axis for mitochondrial biogenesis in hRGCs. We induced mitochondrial damage in hRGCs with CCCP and measured gene expressions of the biogenesis pathway players such as PGC1α, PGC1β, NRF2, NRF1 and PRC. Indeed, we observed significant increases of these gene expressions in both the WT and *E50K* hRGCs upon CCCP-mediated mitochondrial damage (Fig. 3a–d). Next, we measured the biochemical activation of the AMPKα and PGC1α proteins by immunoblotting against

phosphorylated Thr172 and Ser571 for respective proteins under CCCP-induced damage. We found at early CCCP exposure times for both the WT and *E50K* hRGCs, a significant increase in the AMPKα activation which reduced to the basal level during longer treatment (Fig. 3e–g). Furthermore, we observed a moderate activation of the AMPKα effector PGC1α in the WT hRGCs (Fig. 3e, h) but more significant increase in the *E50K* hRGCs under CCCP treatments (Fig. 3e, i). These suggest mitochondrial damage activates the energy sensor AMPKα in hRGCs which leads to the activation of the mitochondrial biogenesis master regulator PGC1α. The stronger PGC1α activation in *E50K* hRGCs suggests the need for more healthy mitochondria, which is supported by our data showing *E50K* hRGCs struggling to restore mitochondrial homeostasis after acute CCCP damage (Fig. 2f).

**TBK1 inhibition promotes mitochondrial biogenesis in hRGCs.** It has been reported that TBK1 inhibits AMPKα in

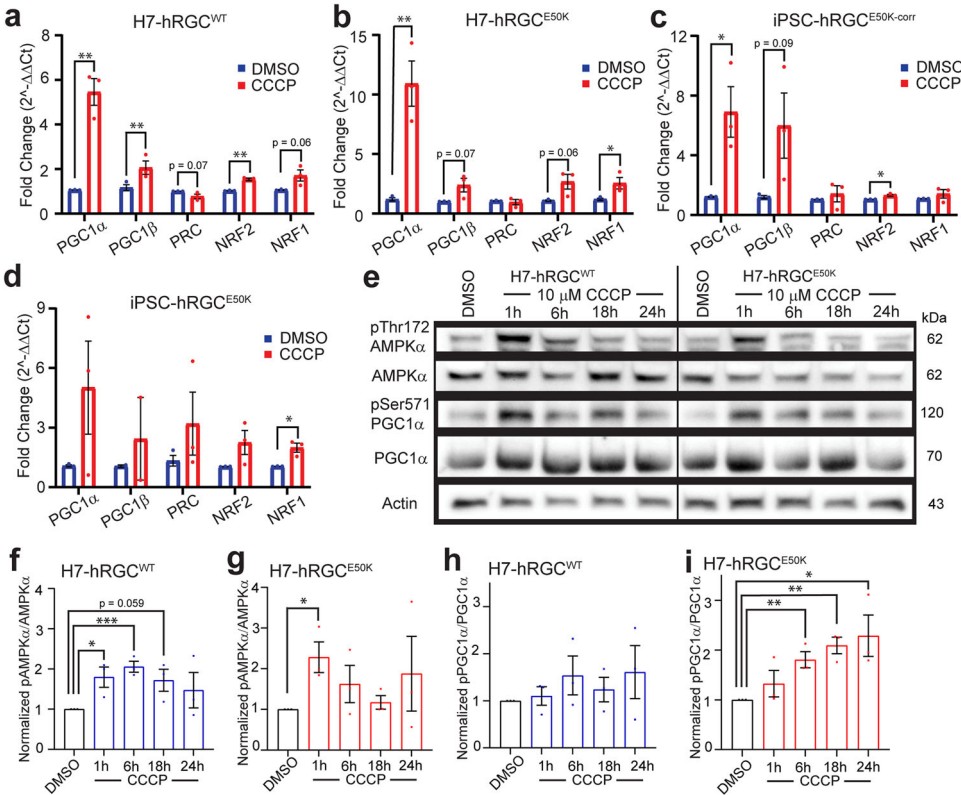

**Fig. 3 Mitochondrial damage activates the AMPKα-PGC1α biogenesis axis in hRGCs.** Mitochondrial biogenesis gene expressions in **a** H7-hRGC$^{WT}$, **b** H7-hRGC$^{E50K}$, **c** iPSC-hRGC$^{E50K-corr}$, and **d** iPSC-hRGC$^{E50K}$ after 24 h CCCP (10 µM) treatment. ΔΔCt fold changes were measured relative to GAPDH, and then DMSO. $n = 3$, 3 technical repeats averaged for each biological repeat. **e** Representative western blot images of H7-hRGCs treated with 10 µM CCCP for the indicated timepoints. **f–i** For each protein, band intensities were quantified and first normalized to corresponding actin loading control, then to its DMSO values, **f**, **g** ratio between p-AMPKα$^{Thr172}$ to AMPKα, and **h**, **i** ratio between p-PGC1α$^{Ser571}$ to PGC1α. $n = 3$. Unpaired student's $t$-test between DMSO and the individual CCCP timepoints. *$p$-value < 0.05, **$p$-value < 0.01, ***$p$-value < 0.001. Error bars are SEM.

adipose tissue, and AMPK activation promotes mitochondrial biogenesis through PGC1α[15]. Hence, we asked if TBK1 inhibition will lead to the AMPK activation which will promote mitochondrial biogenesis in hRGCs. BX795 drug is a potent TBK1 inhibitor[40] that has been shown to resist the nerve fiber layer thinning of the glaucomatous *E50K* transgenic mice retina[41]. We treated both WT and *E50K* hRGCs with BX795 and measured mitochondrial mass by immunofluorescence against the mitochondria-specific Tom20 protein. Surprisingly, we observed at the basal level (DMSO control) *E50K* hRGCs possess significantly lower mitochondrial mass compared to the WT (Fig. 4a, b). To validate that this observation is not due to the indirect effect of DMSO vehicle control[42], we measured mitochondrial mass in untreated WT and *E50K* hRGCs by Tom20 immunofluorescence. Our results further confirmed a significantly lower mitochondrial mass in *E50K* hRGCs compared to the WT (Supplementary Fig. 3a, b). However, TBK1 inhibition by BX795 showed a robust increase in mitochondrial mass at early and longer treatments for the WT and during longer treatments (18 h, 24 h) for *E50K* hRGCs (Fig. 4a, b). We further confirmed the increase in mitochondrial mass by evaluating the mitochondrial DNA copy number under BX795 treatment for both the WT and *E50K* hRGCs (Fig. 4c, d). To investigate whether BX795 increased mitochondrial mass by upregulating the AMPKα-PGC1α axis, we measured the activation status of these proteins by immunoblotting against respective activation site phosphorylation. Interestingly, we observed increased activations of the AMPKα at longer BX795 treatment (24 h) (Fig. 4e, f) and gradual activation of PGC1α for WT hRGCs (Fig. 4e, h). However, such

activation for AMPKα (Fig. 4e, g) and PGC1α (Fig. 4e, i) were not observed for the *E50K* hRGCs under BX795 mediated TBK1 inhibition. Apart from the AMPKα-PGC1α axis, PGC1β is also known to activate mitochondrial biogenesis in cells by activating its target NRF1[43]. Hence, we evaluated PGC1β protein expression in both WT and *E50K* hRGCs, at different BX795 treatment time points. Interestingly, we observed a significant increase in PGC1β level for WT at 18 h while for *E50K* hRGCs the response was quick, and it remained high during 1–18 h BX795 treatment and returned back to the basal level at 24 h of treatment (Supplementary Fig. 4a–c). These data suggest pharmacological inhibition of TBK1 promotes mitochondrial biogenesis in hRGCs, however, for the glaucomatous *E50K* mutation such activation could be independent of the AMPKα-PGC1α pathway.

**Mitochondrial ultrastructure reveals TBK1 inhibition improves mitochondrial coverage and swelling in *E50K* hRGCs.** Our data showed that *E50K* hRGCs possess less mitochondrial mass than the WT neurons (Fig. 4a, b; Supplementary Fig. 3). However, RGCs are highly metabolically active, so it is possible that in *E50K* hRGCs each mitochondrion needs to produce more ATP to meet the metabolic need. Studies have shown that a mild increase of the mitochondrial matrix volume (swelling) can help expand the inner mitochondrial membrane (IMM), which in turn stimulates the electron transport chain activity and helps increase production of ATP[44–46]. Such mitochondrial swelling could be physiological and achieved through the prolonged opening of the permeability transition pores (PTP) in the IMM[47]. We asked if

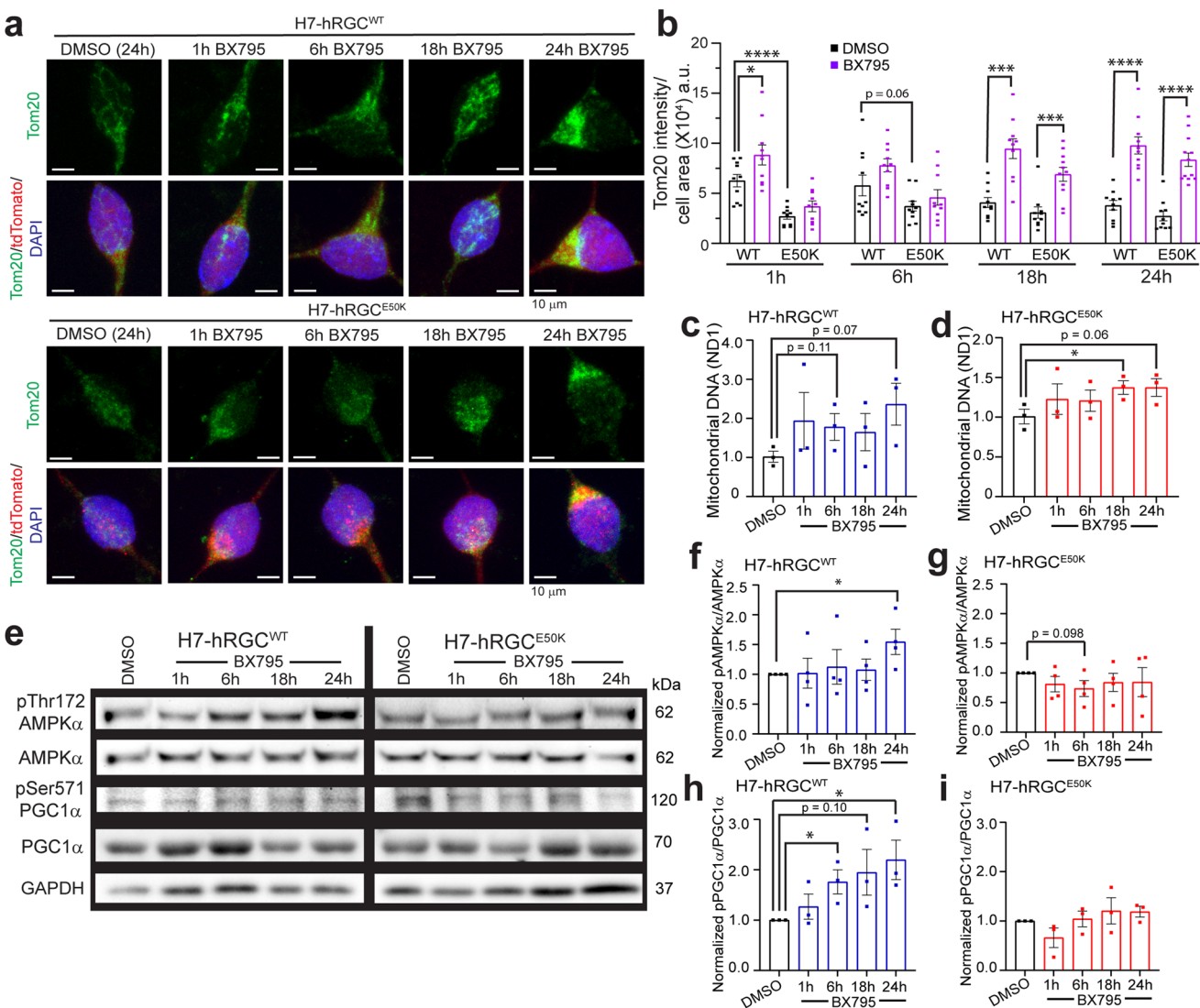

**Fig. 4 TBK1 inhibition activates mitochondrial biogenesis in hRGCs. a** Representative confocal immunofluorescence images of mitochondria (Tom20), tdTomato, and DAPI after 1 µg/ml BX795 or equivalent DMSO treatment for the indicated time. Scale bars are 10 µm. **b** Quantification of Tom20 intensity per cell area from the sum projections of confocal z-stacks. $n = 10$ cells per condition. **c, d** Mitochondrial DNA copy number analyzed by qPCR for mitochondrial ND1 gene normalized to nuclear RNase P gene. Results shown as ΔΔCt fold change normalized to DMSO control for different times points of 1 µg/ml BX795 treatment. $n = 3$, 3 technical repeats averaged for each biological repeat. **e** Representative western blot images of H7-hRGCs treated with 1 µg/ml BX795 for indicated timepoints. **f–i** For each protein, band intensities were quantified and first normalized to corresponding GAPDH loading control, then to its DMSO values, **f, g** ratio between p-AMPKα^Thr172 to AMPKα, and **h, i** ratio between p-PGC1α^Ser571 to PGC1α. $n = 3-4$. *$p$-value < 0.05, **$p$-value < 0.01, ***$p$-value < 0.001, ****$p$-value < 0.0001. Unpaired student's $t$-test between DMSO and the individual BX timepoints. Error bars are SEM.

such a mechanism is in place for the *E50K* hRGCs and performed transmission electron microscopy (TEM) to resolve mitochondrial ultrastructure. As a validation to the lower measured mitochondrial mass in *E50K* hRGCs by immunofluorescence we also observed reduced mitochondrial coverage (mitochondrial section area per unit area of cytoplasm) in glaucomatous *E50K* hRGCs compared to WT hRGCs (Fig. 5a, b). Interestingly, we observed significantly higher perimeter, major, and minor axis lengths for the *E50K* hRGCs in the basal condition (Fig. 5c–e) indicative of swelling which potentially helps mitochondrion to produce more ATP. In support of BX795 mediated activation of mitochondrial biogenesis, TEM analysis further revealed increased mitochondrial coverage in both the WT and *E50K* hRGCs under BX795 treatment (Fig. 5a, b). We asked if increasing mitochondrial mass by BX795 could reduce ATP production load and hence mitigate the swelling in the *E50K* hRGCs. Indeed, we observed a significant reduction in the

mitochondrial perimeter (Fig. 5c), major axis length (Fig. 5d), and a moderate reduction of the minor axis length (Fig. 5e) for *E50K* but not for WT hRGCs in presence of BX795.

**TBK1 inhibition restores energy homeostasis and provides neuroprotection for *E50K* hRGCs.** Our data show that TBK1 inhibition enhances mitochondrial biogenesis and mitigates mitochondria swelling in *E50K* hRGCs. We asked if these improvements are due to the reduced metabolic burden on mitochondria. To test this, we measured cellular metabolisms corresponding to mitochondrial OXPHOS and glycolysis in both WT and glaucomatous *E50K* hRGCs using the Seahorse analyzer under control and BX795 treatments. To measure cellular ATP production rate at the basal level, we performed an ATP rate assay and surprisingly identified *E50K* hRGCs both under control (DMSO) and in presence of BX795 showed higher oxygen

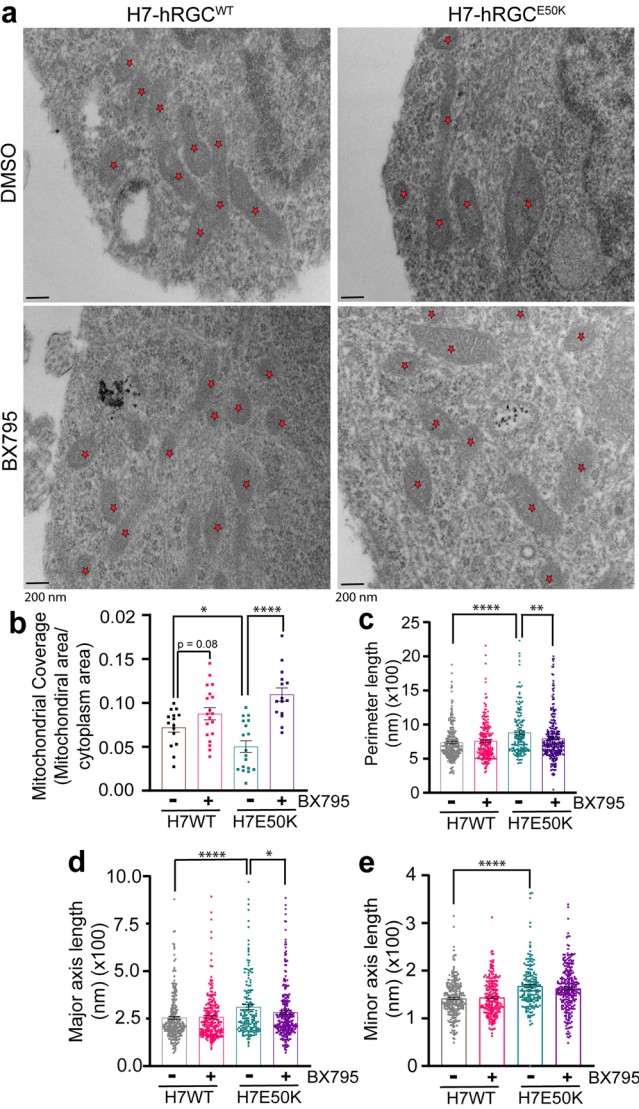

**Fig. 5 TBK1 inhibition improves mitochondrial coverage and mitigates swelling in *E50K* hRGCs. a** TEM images of H7-hRGC$^{WT}$ and H7-hRGC$^{E50K}$ after 24 h DMSO or BX795 (1 μg/ml) treatment, at 49000X magnification. Asterisks represent mitochondria. Scale bar = 200 nm. Quantification of **b** mitochondrial coverage (mitochondrial area/cytoplasm area) for each image, **c** perimeter length, **d** major axis length, and **e** minor axis length for each mitochondrion. **b** Unpaired student's *t*-test or **c–e** Mann–Whitney *U* test between independent datasets, **p*-value < 0.05, ***p*-value < 0.01, *****p*-value < 0.0001, *n* = 15–24. Error bars are SEM.

consumption rate (OCR) than that of WT; however, BX795 treatment itself did not change OCR (Fig. 6a). Correspondingly, we observed increased total and mitoATP production rates in *E50K* hRGCs compared to the WT under control and BX795 treatments (Fig. 6b). It has been reported that cellular stress could activate oxidative phosphorylation and increase OCR[48], suggesting increased cellular stress in the glaucomatous *E50K* hRGCs. Interestingly, we also observed increased glycoATP production rate in the *E50K* hRGCs compared to WT under BX795 treatment (Fig. 6b), presumably due to the high OCR and activation of aerobic glycolysis which is often observed in a variety of cell types[49]. BX795 increased mitochondrial mass in the WT and *E50K* hRGCs, but it did not alter OCR compared to the DMSO control (Fig. 6a). This observation could be supported if the consumed oxygen (OCR) is distributed among an increased

number of mitochondria, which will lead to reduced mitoOCR under BX795 treatment to maintain energy homeostasis. To test this, we performed a glycolytic rate assay where serial inhibition of mitochondrial activity by rotenone plus antimycin A results in decreased OCR (Fig. 6c) and glycolysis inhibition by glucose analogue 2-deoxy-d-glucose (2-DG) results in reduced proton efflux rate (PER) (Fig. 6d). These parameters are used to extract mitoOCR and glycoPER for their respective contributions. In support of the higher mitoATP production rate in the ATP rate assay (Fig. 6b) we also observed increased mitoOCR/glycoPER in the *E50K* hRGCs compared to the WT cells at the basal level (Fig. 6e). Furthermore, we also observed reduced mitoOCR/glycoPER in the *E50K* hRGCs under BX795 treatment (Fig. 6e). This supports the notion that increased mitochondrial mass lowers mitoOCR to maintain energy homeostasis. This data is further supported by the observation that in *E50K* hRGCs mitochondrial swelling reduced under BX795 treatment (Fig. 5c–e), presumably due to the lower ATP production load per mitochondrion. We next performed a Mito Stress Test to study if increased mitochondrial mass under BX795 treatment helps to combat degenerative conditions as an indication for neuroprotection. In agreement to the above results, at the basal level BX795 treatment maintained similar OCR for both the WT (Fig. 6f) and *E50K* hRGCs (Fig. 6g). However, under FCCP induced maximum respiration, we observed increased OCR under BX795 treatment for both the WT and *E50K* hRGCs (Fig. 6f, g). This led to an increased spare respiratory capacity (difference between maximal and basal respiration) for both WT and glaucomatous hRGCs when treated with BX795 (Fig. 6h). An increase in spare respiratory capacity reflects cells' ability to produce additional ATP by OXPHOS to combat degenerative conditions[50]. This suggests TBK1 inhibition-mediated activation of mitochondrial biogenesis could confer neuroprotection to the ganglion cells.

RGCs are progressively degenerating neurons which exacerbates during aging and under disease conditions[6]. Apoptotic RGC death is the hallmark for glaucoma irrespective of the causal factors[51]. To test if BX795 promotes a pro-survival effect, we measured apoptosis master regulator Caspase-3/7 activity in the WT and *E50K* hRGCs under basal and CCCP-induced damage. Remarkably, we observed 24 h of BX795 treatment reduced Caspase activity for the WT and *E50K* hRGCs both under basal and CCCP damage (Fig. 6i, k). We did not observe any increase in Caspase activity for the WT hRGCs even after 48 h of CCCP treatment with no additional effect of BX795 (Fig. 6j) presumably due to the existence of stringent MQC mechanisms. However, glaucomatous *E50K* hRGCs showed a strong increase in caspase activity post 48 h of CCCP treatment which is mitigated by the BX795 treatment (Fig. 6l) as evidence for neuroprotection. Furthermore, 48 h of BX795 also reduced basal Caspase activity in *E50K* hRGCs (Fig. 6l) as an indication for sustained neuroprotection under glaucomatous conditions.

Neurodegenerative diseases are often associated with the protein aggregation[52] and it has been reported that in cultured HEK293 cells OPTN$^{E50K}$ protein forms insoluble aggregates which were dissolved by BX795[53]. Crystal structure analysis revealed the OPTN$^{E50K}$ protein forms tight electrostatic interactions with the E698 residue of TBK1 protein resulting in insoluble aggregates[53,54]. However, the mechanism of BX795-mediated OPTN$^{E50K}$ aggregate dissolution is yet to be discovered. We asked if OPTN$^{E50K}$ forms these insoluble aggregates in hRGCs, and if they could be dissolved by inhibiting TBK1 with BX795. Indeed, by confocal immunofluorescence we observed larger OPTN aggregates in the *E50K* hRGCs compared to WT cells, which were then dissolved by TBK1 inhibition using BX795 (Supplementary Fig. 5a, b). Thus, our finding of reduced cellular apoptosis in the *E50K* hRGCs under BX795 treatment may be a

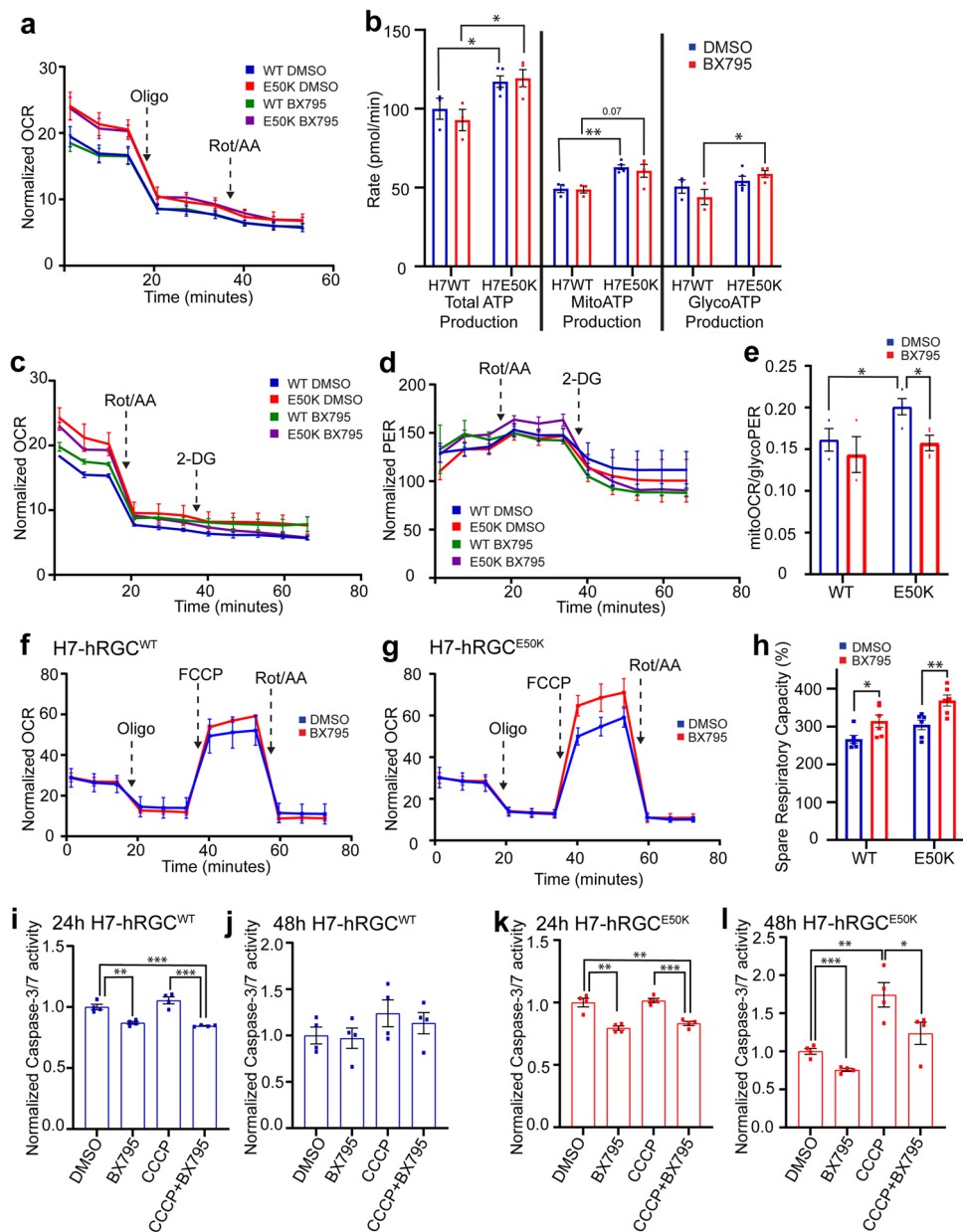

**Fig. 6 BX795 mediated TBK1 inhibition restores energy homeostasis and provides neuroprotection in glaucomatous hRGCs. a** Measurements of cell area normalized OCR from the ATP rate assay of hRGCs treated with DMSO or 1 μg/ml BX795 for 24 h. **b** Calculated total-, mito-, and glyco- ATP production rates from the ATP rate assay. $n = 3$–5. **c**, **d** Measurements of cell area normalized **c** OCR and **d** PER from the glycolytic rate assay. **e** Calculated mitoOCR/glycoPER from the glycolytic rate assay. $n = 3$–4. **f**, **g** Measurements of cell area normalized OCR from the Mito Stress Test of **f** H7-hRGC^WT and **g** H7-hRGC^E50K, and **h** spare respiratory capacity after 24 h treatment with 1 μg/ml BX795. $n = 6$. **i–l** Caspase-3/7 activity was measured using ApoToxGlo-Triplex assay kit in hRGCs after 16 h pretreatment with BX795 and then 24 h or 48 h treatment with DMSO, 1 μg/ml BX, 10 μM CCCP, or BX795 and CCCP treatments. Treatment conditions were normalized to the DMSO control for respective hRGC type. *$p$-value < 0.05, **$p$-value < 0.01, ***$p$-value < 0.001. Unpaired student's $t$-test between independent datasets. Error bars are SEM.

combined effect of restoring energy homeostasis and aggregate dissolution.

## Discussion

In this study, we have identified mitochondrial biogenesis as a critical step of MQC which can be activated to promote cell survival for both the WT and glaucomatous *E50K* RGCs. RGCs are highly sensitive towards mitochondrial damage as they need a steady source of ATP for spontaneous action potential firing[55]. Mitochondrial damage potentially disrupts ATP supply and triggers an energy sensing pathway through AMPK[35]. We have

shown here acute damage to mitochondria activates mitochondrial biogenesis in RGCs with robust clearance of damaged mitochondria to maintain mitochondrial homeostasis and energy supply. Activation of this biogenesis occurs through the AMPK-PGC1α biogenesis axis. Previously, it was reported that the mitochondrial biogenesis activation could be pro-survival as it associates with exercise-induced muscle growth[33] and synaptic plasiticity[34]. In support to this notion, we found a robust increase of mitochondrial biogenesis in hRGCs with minimal cell death under mitochondrial stress as we previously reported[23].

Degradation of damaged mitochondria and biogenesis of healthy organelle are central to the maintenance of energy

homeostasis, which is critical for neuronal health. AMPK functions as an energy sensor and responds to energy stress by activating mitochondrial biogenesis[35,56,57]. It has been shown in mice adipose tissue that TBK1 inhibits AMPK activity causing suppression of energy expenditure, obesity, and inflammation[15]. TBK1 is also a key effector for cellular inflammation in response to pro-inflammatory stimuli, and TBK1 inhibition leads to reduced obesity, inflammation, insulin resistance, and more energy expenditure in mice in response to a high-fat diet[15]. There are several AMPK activators currently under clinical trials for metabolic diseases[58]. Among them metformin is widely used for type 2 diabetes, however the drug functions through inhibiting mitochondrial complex I which reduces ATP production[58]. Our study provides an indirect activation mechanism of AMPK by targeting TBK1 without compromising mitochondrial function. Thus, TBK1 provides a very interesting target for developing neuroprotection through restoring energy homeostasis which remains unexplored for CNS disorders. We have shown here that TBK1 inhibition by BX795 leads to increased mitochondrial biogenesis in both the WT and glaucomatous (E50K) hRGCs. Interestingly, in WT hRGCs, but not in E50K hRGCs, the above treatment activated the AMPKα-PGC1α mito-biogenesis signaling axis. However, PGC1β protein level increased in both the WT and E50K hRGCs under BX795 treatment suggesting the existence of a PGC1α independent biogenesis activation pathway. It has also been reported that mitochondrial biogenesis could be activated independent of AMPK and PGC1α[59,60], however further studies will be required to delineate the mechanisms of TBK1 inhibition-mediated activation of mitochondrial biogenesis in the E50K hRGCs. Increased mitochondrial mass by BX795 restored energy homeostasis, reduced mitochondrial swelling, and enhanced spare respiratory capacity in the E50K hRGCs as an indication for neuroprotection. Remarkably, BX795 treatment not only reduced cellular apoptosis at the basal level but also under CCCP-induced apoptosis in the E50K hRGCs, making it a strong candidate for OPTN mutation-associated glaucoma neuroprotection therapy. It is intriguing to see that BX795 treatment to the WT hRGCs also increased spare respiratory capacity and reduced cellular apoptosis both under basal and CCCP-induced acute mitochondrial damage. Activation of pro-survival pathways under basal conditions in the WT hRGCs in response to TBK1 inhibition suggests these neurons are not in a homeostatic condition. However, it is known RGCs are spontaneously degenerating neurons with reduced mitochondrial function[6,61]. Thus, these findings become more important particularly for developing RGC protection for optic neuropathies where no obvious genetic disorders are associated, such as during aging and in high intraocular pressure-induced glaucoma. Of note, TBK1 copy number increase mutations are found among the normal-tension glaucoma patients[62]. This may lead to reduced mitochondrial biogenesis and disrupted energy homeostasis-related RGC death. Our study provides a therapeutic strategy where TBK1 inhibition could promote mitochondrial homeostasis-mediated RGC neuroprotection for the above glaucoma conditions.

Conflicting evidence suggests TBK1 inhibition could activate[15,63] or inhibit[64] cellular inflammation and activation of inflammation can lead to cell death. As TBK1 inhibition resulted in reduced apoptosis in hRGCs, it is unlikely to activate inflammation. TBK1 activity is also critical for activating mitophagy proteins such as OPTN, NDP52, TAX1BP1 and P62 for successful mitophagy[65] and defects in mitophagy could lead to neurodegeneration[66]. However, it has also been reported that activation of the above mitophagy proteins could occur independent of TBK1, but through ULK1 kinase[67]. Thus, the neuroprotective effect of TBK1 inhibition is unlikely to cause mitophagy defects.

Mitochondrial dysfunction is the root cause for optic neuropathies such as Leber's hereditary optic neuropathy, dominant optic atrophy[2] and some cases of glaucoma[10]. Our study provides a neuroprotection strategy for a broad range of optic neuropathies irrespective of the genetic disorders. Severe mitochondrial abnormality has been reported for dopaminergic neurons[3] and motor neurons[4] in Parkinson's disease and amyotrophic lateral sclerosis respectively. Our study could be applied to test if TBK1 inhibition leads to increased mitochondrial homeostasis and neuroprotection for these CNS disorders.

## Methods

**Reagents and resources**. All the reagents including qPCR primers, antibodies, and software used are listed in Supplementary Table 1

**ESC maintenance**. Human embryonic stem cell (H7-hESCs; WiCell, Madison, WI, https://www.wicell.org) reporter line with CRISPR-engineered multicistronic BRN3B-P2A-tdTomato-P2A-Thy1.2 construct into the endogenous RGC specific BRN3B locus was used for isogenic control[20]. CRISPR mutated H7-hESC reporter with OPTN$^{E50K-homozygous}$ mutation (H7-E50K) was done as explained previously[25]. Patient-derived induced pluripotent stem cells (iPSCs) with E50K mutation[24] (iPSC-E50K), E50K mutation corrected to WT by CRISPR in the patient-derived iPSC (iPSC-E50Kcorr)[25] with BRN3B::tdTomato reporter were obtained from the Jason Meyer lab. All the above cell lines were grown in mTeSR1 media (mT) at 37 °C in 5% CO$_2$ incubator on matrigel (MG) coated plates. These cells were maintained by clump passaging using Gentle Cell Dissociation Reagent (GD) after 70–80% confluency. Media was aspirated and GD was added to cells followed by incubation at 37 °C in 5% CO$_2$ incubator for 5 min. Next, mT was used to break up the colonies into small clumps by repeated pipetting and then seeded onto new MG plates.

**hRGC differentiation and immunopurification**. For differentiation, stem cells were dissociated to single cells using accutase for 10 min and then quenched with twice the volume of mT with 5 μM blebbistatin (blebb). The cells were centrifuged at 150 xg for 6 minutes and resuspended in mT with 5 μM blebb, then 100,000 cells were seeded per well of a 24-well MG coated plate. The next day, media was replaced with mT without blebb. After 24 h, media was replaced with differentiation media (iNS) and further small molecule-based differentiation was carried out with iNS media as elucidated previously[23]. Differentiation of hRGCs was monitored by tdTomato expression and cells were purified during days 45–55 using THY1.2 microbeads and magnetic activated cell sorting system (MACS, Miltenyi Biotec) as explained before[20,23]. Next, hRGCs were resuspended in iNS media, counted using a hemocytometer and seeded on MG-coated plates, coverslips, or MatTek dishes for experiments.

**Flow cytometry**. Purified hRGCs were seeded at 30,000 cells per well of a 96-well MG-coated plate and maintained for 3 days. For measuring mitochondrial mass, hRGCs were labeled with mitochondria-specific MTDR dye. To measure mitochondrial degradation, hRGCs were labeled with 10 nM MTDR dye for 1 h, washed, and then treated with 10 μM CCCP for a time course. To measure mitochondrial biogenesis, hRGCs were labeled with MTDR first, treated with 10 μM CCCP or equal amount of DMSO for 3 h, then washed and incubated with fresh media with MTDR for the time course. After treatments, hRGCs were dissociated to single-cell suspension using accutase and analyzed using Attune NxT flow cytometer (Thermo Fisher).

**Western blot**. Purified hRGCs were seeded at 500,000 cells per well of 24-well MG-coated plates and maintained for 3 days. Cells were then treated with indicated molecules and time points. DMSO was used as the control as the small molecules were dissolved in DMSO. The cells were lysed and collected in 100 μl of M-PER extraction buffer with 5 mM EDTA and protease inhibitors. Protein quantification was done using a BSA standard following the DC Protein Assay Kit II (Bio-Rad) and measured on microplate reader. Loading samples were prepared with heat denaturation at 95 °C for 5 minutes with laemmli sample buffer (1X). 10–20 μg protein per sample were run on Bio-Rad Mini-PROTEAN TGX precast gels, in running buffer (Tris-Glycine-SDS buffer; Bio-Rad) at 100 V until the dye front reached to the bottom. The transfer sandwich was made using PVDF membrane (activated in methanol) in Tris-Glycine transfer buffer (Bio-Rad) with 20% methanol and transferred for 2 h at 30 V, 4 °C. For the visualization of proteins, the membranes were blocked in 5% skim milk in TBST (TBS buffer with 20% Tween20) for 2 h at room temperature and incubated overnight at 4 °C in 1:1000 dilution of primary antibodies for PGC1α (Abcam), Phospho-PGC1α$^{Ser571}$ (R&D Systems), PGC1β (Abcam), AMPKα (Cell Signaling Technologies, CST), Phospho-AMPKα$^{Thr172}$ (CST), LC3B (Sigma), GAPDH (CST), or ACTIN (CST). Membranes were then washed three times for 5 minutes each in TBST, followed by 2 h of incubation in 5% milk in TBST containing anti-rabbit HRP linked secondary

antibody (CST) at 1:10,000 dilution. The membranes were again washed three times with TBST and then placed in Clarity Max Western ECL (Bio-Rad) substrate for 5 min. The membranes were imaged in a Bio-Rad ChemiDoc Gel Imager, and the raw integrated density for each band was measured and normalized with respect to the corresponding GAPDH or actin loading control using Image J. Treatment conditions were further normalized to the corresponding DMSO control for each experiment. Complete blots of the representative western blot images, with protein molecular weight marker (Thermo Scientific), are provided in Supplementary Fig. 6. Protein bands corresponding to the appropriate size were quantified following product datasheets and published literature.

**Immunofluorescence and Confocal imaging**. Purified hRGCs were seeded on MG-coated coverslips (1.5 thickness) at a density of 30,000–40,000 cells per coverslip and grown for 3 days. Next, hRGCs were treated with indicated molecules and time points. After treatment the media was aspirated, the cells were washed with 1× PBS, and then fixed with 4% Paraformaldehyde for 15 min at 37 °C. Fixed cells were permeabilized with 0.5% Triton-X100 in PBS for 5 min and then washed with washing buffer (1% donkey serum, 0.05% Triton-X100 in PBS) three times for 5 min each. Cells were blocked with blocking buffer (5% donkey serum, 0.2% Triton-X100 in PBS) for 1 h. After blocking, antibodies against TOM20 (Mouse, Santa Cruz), Tubulin β3 (mouse, Biolegend), RBPMS (rabbit, GeneTex) and Optineurin (Rabbit, Cayman Chemicals) were added (1:200 in blocking buffer) and the coverslips were incubated overnight at 4 °C. Next, coverslips were washed with washing buffer three times for 5 min each and incubated for 2 h at room temperature in the dark with fluorophore conjugated anti mouse or rabbit secondary antibodies (1:500). The coverslips were washed with washing buffer three times for 5 min each, with 1.43 μM DAPI added to the second wash. The coverslips were then mounted onto slides with DAKO mounting medium. Visualization of above proteins and nucleus was done by confocal immunofluorescence microscopy using Zeiss LSM700 with 63x/1.4 oil objective. Analysis was carried out using ImageJ with maximum projections of DAPI channel (number of nuclei) and sum projections of TOM20 and OPTN channels for the corresponding confocal z-stacks. For OPTN aggregate size, we analyzed particles from 0.02 a.u. to infinity to account for the small and big aggregates.

**qPCR**. Purified hRGCs were seeded at 300,000 cells per well of 24-well MG-coated plates and maintained for 3 days. Cells were then treated with indicated molecules and durations. Media was aspirated and cells were incubated in 200 μl accutase for 10 min and then quenched with 400 μl iNS media. Cells were centrifuged at 150 xg for 6 min, media aspirated, and the cell pellets were stored at −20 °C. RNA was extracted from cell pellets following the kit protocol (Qiagen 74104). The RNA concentration was measured using Nanodrop 2000c (Thermo) and 6 μl of RNA was used to prepare cDNA following the kit protocol (Abm #G492). Primers were designed as detailed in Table S1 and qPCR were performed using BlasTaq qPCR MasterMix with 100 ng total cDNA in a 20 μl reaction mixture using QuantStudio6 Flex RT PCR system (Applied Biosystems). GAPDH or actin was used as a housekeeping gene in every plate to calculate the ΔCt values. The ΔΔCt was calculated with respect to (w.r.t) the average ΔCt of the control sample. All conditions were measured by averaging three technical repeats for each biological repeat with total three biological replicates.

**Mitochondrial DNA measurements by qPCR**. Purified hRGCs were seeded at 50,000–75,000 cells per well of 96-well MG-coated plates and maintained for 3 days. Cells were then treated with indicated molecules for the designated durations. Media was aspirated and cells were incubated in 30 μl accutase for 10 min and then quenched with 100 μl iNS media. Cells were centrifuged at 150 xg for 6 min, media aspirated, and the cell pellets were stored at −20 °C. DNA was extracted using DNeasy Blood & Tissue Kit (Qiagen) and eluted with 30 μl elution buffer. DNA concentration was measured using Nanodrop 2000c (Thermo). 10 ng of DNA was used to measure both mitochondrial ND1 gene and internal control, human nuclear RNase P gene copy numbers, as done previously[23].

**JC1 imaging**. hRGCs were seeded on MG-coated glass bottom (1.5 thickness) MatTek dishes at 40,000 cells per dish and maintained for 3 days. For the experiments in Fig. 2a–c, cells were incubated with 250 μl of JC1 media (1:100 in iNS) for 30 min in the incubator, then an additional 1.75 ml of dilute JC1 (1:1000 in iNS) was added to the dish. The dish was then transferred to the live cell chamber (5% CO₂, 37 °C, Tokai Hit) and confocal z-stacks were acquired prior to (before) and 10 minutes after CCCP (10 μM) treatment using Zeiss LSM700 with 63x/1.4 oil objective. For experiments in Fig. 2d–f, cells were treated with 10 μM CCCP or equivalent DMSO for 30 min. Next, cells were washed with iNS media, and then incubated with 250 μl of JC1 media (1:100 in iNS) for 30 min in the incubator. The JC1 media was then removed, cells were washed again, and 2 ml of new iNS media was added to the dish. The dish was then transferred to the live cell chamber and imaged. Analysis was carried out using ImageJ with sum projections of red and green channels. The red fluorescence from the tdTomato expressed by the hRGCs was much less intense than the JC1 red staining of mitochondria, thus the red fluorescence measured from the cytoplasm was considered background and subtracted out of the measurements. For the green fluorescence, background was

measured from outside the cell. Red-to-Green ratios were calculated by dividing the red intensity by green intensity. These values were then normalized to the control condition, hRGC^WT-before (Fig. 2c) or average DMSO ratio for each cell line (Fig. 2f).

**Seahorse analysis**. The 96-well seahorse plate was coated with MG and hRGCs were seeded at 250,000 cells per well and maintained for 2 days. 24 h before measurements, media was exchanged with 100 μl iNS with 1 μg/ml BX795 or equivalent vehicle control DMSO. A day prior to the assay, the sensor cartridge was fully submerged with 200 μl of sterile water to hydrate it overnight in a non-CO₂, 37 °C incubator. The next day, sterile water was replaced with pre-warmed XF calibrant buffer (Agilent) and the sensors were again submerged and incubated in the non-CO₂, 37 °C incubator for 45–60 min. Seahorse media was made by adding stock solutions to XF DMEM to have final concentrations of 21.25 mM glucose, 0.36 mM sodium pyruvate, and 1.25 mM L-glutamine (Agilent), with pH adjusted to 7.38–7.42. Depending upon the assay, 20 μM Oligomycin (Oligo), 20 μM FCCP, 2.5 μg/ml Rotenone plus 5 μM Antimycin A (Rot/AA), and/or 175 mM 2-deoxy-d-glucose (2-DG) solutions were then prepared in Seahorse media. In the Seahorse plate with hRGCs, iNS media was carefully exchanged to Seahorse media by removing, 60 μl iNS from all wells and adding 140 μl of Seahorse media. Next, 140 μl of the mixed media was removed by pipette and then an additional 140 μl seahorse media was added to each well to have a final volume of 180 μl. 180 μl of Seahorse media was then added to any empty wells. The plate was placed in Incucyte S3 (Sartorius) and one image of each well was taken for cell area normalization using brightfield and red fluorescence (tdTomato) channels. The hRGC seahorse plate was then placed into a non-CO₂, 37 °C incubator for at least 45 min. The reagents were added into their respective ports in the cartridge with the final concentrations in the wells as follows. For ATP rate assay, 2.0 μM Oligo and 0.25 μg/ml Rotenone plus 0.5 μM Antimycin A; for the glycolytic rate assay, 0.25 μg/ml Rotenone plus 0.5 μM Antimycin A and 17.5 mM 2-DG; and for the Mito stress test, 2.0 μM Oligo, 2.0 μM FCCP (optimal concentration from FCCP titration), and 0.25 μg/ml Rotenone plus 0.5 μM Antimycin A. After loading all ports, the cartridge was placed into a non-CO₂, 37 °C incubator while the experiment was setup in WAVE software (Agilent). The cartridge was then placed into the XFe96 analyzer (Agilent) and run for calibration. After the machine calibrations were successful, the hRGC seahorse plate was placed into the machine and the assay was run (ATP rate assay, glycolytic rate assay, or Mito stress test). Cell area from Incucyte images were measured using Image J and extrapolated for the total cell area in each well for normalization. The assay results were then exported into the appropriate excel macro using Seahorse Wave Desktop software (Agilent) for analysis.

**Caspase activity**. hRGCs were seeded at 25,000 cells per well of a 96-well clear bottom black-walled plate and maintained for 3 days. The cells were then treated with indicated molecules for the designated time points. The caspase activity was measured using the ApoTox-Glo Triplex assay kit (Promega). 100 μl of Caspase-Glo® 3/7 reagent was added to all wells and incubated for 30 minutes at room temperature before measuring luminescence (Caspase). Measurements were normalized to DMSO control.

**Electron microscopy**. hRGCs were seeded at 250,000 cells per well on MG-coated 6-well plates and maintained for 3 days. The cells were then treated with 1 μg/ml BX795 or equivalent DMSO for 24 h. Media was aspirated, 500 μl of fixative solution (3% Glutaraldehyde, 0.1 M Cacodylate) was added, and the cells were incubated for 15 min. Next, hRGCs were scraped and pelleted by centrifugation at 10,000 xg for 20 minutes. The pellets were fixed overnight at 4 °C, then rinsed the next day in 0.1 M cacodylate buffer, followed by post fixation with 1% osmium tetroxide, 0.1 M cacodylate buffer for 1 h. After rinsing again with 0.1 M cacodylate buffer, the cell pellets were dehydrated through a series of graded ethyl alcohols from 70 to 100%, and 2 changes of 100% acetone. The cell pellets were then infiltrated with a 50:50 mixture of acetone and embedding resin (Embed 812, Electron Microscopy Sciences, Hatfield, PA) for 72 h. Specimen vial lids were then removed, and acetone allowed to evaporate off for 3 h. Then the pellets were embedded in a fresh change of 100% embedding resin. Following polymerization overnight at 60 °C the blocks were then ready for sectioning. All procedures were done in centrifuge tubes including the final embedding. Sections were cut at 85 nm, placed on copper grids, stained with saturated uranyl acetate, viewed, and imaged on a Tecnai Spirit (ThermoFisher, Hillsboro, OR) equipped with an AMT CCD camera (Advanced Microscopy Techniques, Danvers, MA). 49000X images were analyzed using Image J to measure mitochondrial parameters as explained in Fig. 5.

**Statistics and reproducibility**. Samples were treated with CCCP or BX795 at different time points as independent biological samples. Statistical tests between two independent datasets were done by Student's t-test, each data point within a dataset is from an independent culture well or cell (Figs.1c–f, h, 1m; j, l, 2c, f, 3a–d, f–i, 4b–d, f–i, 5b, 6b, e, h–l, Supplementary Figs. 3b, 4b–c). We used t-test rather than ANOVA because we did not want to assume that each group has the same variance. For non-normal data distribution, we performed Mann–Whitney U test to compare between two independent data sets (Fig. 5c–e, Supplementary Fig. 5b).

Graphs were made using GraphPad Prism 9.0 software and figures were made using Adobe Illustrator.

**Reporting summary**. Further information on research design is available in the Nature Portfolio Reporting Summary linked to this article.

## Data availability

The data used to create all figure graphs is provided in Supplementary Data 1. All uncropped images of western blots are presented in Supplementary Fig. 6. All cell lines and data used during the current study are available from the corresponding author upon reasonable request.

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

## Acknowledgements

We thank the Broxmeyer laboratory and the Hypoxia Core Facility of the Indiana University Cooperative Center for Excellence in Hematology (U54 DK106846) for their helpful discussion and assistance in Seahorse experiments; D. Zack for valuable discussions and H7-hESC reporter line; C. Miller from Electron Microscopy Core for sample processing and TEM imaging; and G. Eckert for providing consultations on statistical analysis. This project was funded by grants from the National Institutes of Health, USA (R00EY028223 to A.D. and R01EY033022 to J.M.). This work was supported in part by a Challenge Grant from Research to Prevent Blindness to the Department of Ophthalmology, Indiana University School of Medicine, by the Indiana Clinical and Translational Sciences Institute; by an award from The Glaucoma Foundation (TGF) to A.D., by an award from Ralph W. and Grace M. Showalter Research Trust and the Indiana University School of Medicine to A.D. and by BrightFocus foundation (G2020369 to J.M.). The work was funded in part by the Paul and Carole Stark Fellowship and Grants in aid of research from Sigma Xi - The Scientific Research Honor Society, to the graduate student K.A., by the Short-Term Training in Ophthalmology Research for Medical Students funded, in part by T35EY031282 from the National Institutes of Health. The content is solely the responsibility of the authors and does not necessarily represent the official views of the National Institutes of Health.

## Author contributions

A.D. conceived and supervised the project. A.D., M.S., K.A., S.D., L.J.O. designed and performed experiments and analyzed data. K.C.H. and J.M. developed the *E50K* reporter lines and contributed to experimental designs. M.S., K.A., and A.D wrote the manuscript.

## Competing interests

The authors declare no competing interests.
