## [Peer Review File · Communications Biology]

Reviewers' comments:

Reviewer #1 (Remarks to the Author):

Dear Authors,

Your manuscript on mitochondrial biogenesis in cultured human RGCs represents a well-written account of a well-conducted study. I have a few comments and suggestions for you to consider, as listed below.

One general comment first. You state in your discussion that AMPK is a 'great' drug target. I would maybe say it represent an 'interesting' potential drug target. Since this seems to be a key take-away message from your paper, please consider if a more elaborate discussion on this would add to your manuscript. I know there are phase II/III studies on the way with direct AMPK activators and that indirect activators (metformin) have been in clinical use for decades and are being repurposed for e.g. cancer. There are some interesting differences to consider in the context of your work since you use a drug that indirectly activates AMPK but do not inhibit mitochondrial respiration/ATP production as I assume metformin does.

Comments to the analysis and presentation of the data, figure by figure:

1. Fig. 1: Is it reasonable to infer that no difference in Tom70/Tom20 expression among the cell lines are indicative of no difference in total mitochondrial mass? Has this been shown experimentally? Please cite relevant studies.
2. Fig. 1: Please indicate the SD for each datapoint presented in panels b and c.
3. Fig. 1, panels b and c: The mass stayed similar, you state. But the level of TOM70 seem to be dynamic and no statistical analysis is offered. Please explain and revise the manuscript to clarify this point.
4. Fig. 1: At this point in the manuscript, the rationale for choosing 10 μ M CCCP is not clear to me. How was the concentration chosen and what was the goal?
5. Fig. 1, panel f: I'm confused here. Using the MTDR dye you say that mitochondrial degradation is increased by CCCP. Is that not the same as mitochondrial mass is decreasing? Please explain and revise the manuscript to clarify this point. Please link your reply to comments 1 and 3 above. Please also explain the choice of statistical analysis. I would tend to think a two-way/repeated measures ANOVA would be more appropriate here since you have two dependent variables.
6. Fig. 1, panel g: Please back up the statement regarding an increase in the MTDR signal to baseline levels with a statistical analysis (two-way/repeated measures ANOVA perhaps?)
7. Fig. 1 and other figures with WB data: Please show the entire blots in the supplementary section including the MW markers.
8. Fig. 2: As a general note for fig. 2 and other figures, I would like to applaud you for showing the individual data points. This is something that needs to become the standard way of showing data for what I think are obvious reasons.
9. Fig. 2: In relation to the two-way ANOVAs that you do, it would be interesting to see the analysis in the contribution of the two dependent variables to the total variation in the dataset. Please consider presenting this analysis.
10. Fig. 3: As suggested for the plots in fig. 1, please consider showing the error bars in panels f-I and

do a repeated measures ANOVA.

11. Fig. 4: Again, nicely layed out figure. I will just repeat myself here and ask that you consider a statistical analysis for the data presented in panels f and h-l.

12. Fig. 5: For this figure you chose to present the data differently employing what looks like violin plots in panel g. For panels e and f, I assume the vertical line represents the mean, but this is not stated. Please elaborate on the choice of data visualization for this figure.

13. Fig. 6 and suppl. Figures, no comments.

Reviewer #2 (Remarks to the Author):

In this manuscript, Surma et al aim to show that human retinal ganglion cells (hRGCs) efficiently generate mitochondria under mitochondrial damage conditions, and to demonstrate that TBK1 inhibition activates mitochondrial biogenesis and exerts neuroprotective effects in both WT and glaucomatous RGC neurons by pharmacological inhibition of TBK1 or activation of AMPK for mitochondrial production and cell survival. Although the current study is potentially interesting, it suffers from significant data inconsistency and lack of in-depth investigation e.g., upon mitochondrial clearance and TBK inhibition.

Specific comments are appended:

1. Figure panels should be better arranged. Currently data quantification is separated from figure data. For example, Fig 1a (top two panels) and 1b should be labelled as Fig 1a; Fig 1a (bottom two panels) and Fig 1c should be labelled as Fig 1b. Such changes should be made throughout the manuscript.
2. In Fig 2, how does mitochondrial clearance occur in the hRGCs under CCCP treatment? It is necessary to provide direct evident to support this. Mitophagy should be tested here.
3. It seems that mitochondrial (JC1 red) levels in E50K cells are not reduced, compared with WT cells (Fig 2b-d). This raises a question whether E50K mutation is relevant to the following experiments.
4. It is not clear how the treatment was done in Fig 4b, and the Fig legend does not correspond to the figure.
5. Fig 4a-b, why could OPTN aggregates be dissolved in the E50K-hRGCs with TBK1 inhibition?
6. WT and E50K cells appear respond to TBK1 inhibition differently in Fig 4c-d, but in Fig 4e-f, the two types of cells behave similarly. How could this be the case?
7. The immunoblot data in Fig 4e, g does not appear convincing. The increase in Tom70 levels seems transient, and the enhanced levels in PGC-1a, b or pAMPK/AMPK are modest or transient.
8. In Fig 5, BX795 treatment seems to give rise to significant effects on mitochondrial levels in E50K cells, but not in WT cells. This is somewhat confusing, and inconsistent with the data in Fig 4 and Fig 6.
9. In Fig 6, BX795 treatment appears to promote the survival for both WT and E50K hRGCs, but not for the iPSC cells, however it reduces caspase activity in iPSC cells.

Minor points:

Page numbers should be added.

What does w.r.t. stand?

Reviewer #3 (Remarks to the Author):

General comments

1. The authors have to provide the justification for the use of 10uM of CCCP for treatment. It is unclear why and how these conditions were selected.
2. The quality of mitochondrial staining using fluorescence dyes is unacceptable. Images throughout the manuscript are exceptionally poor not allowing to visualize individual mitochondria.
3. CCCP is an uncoupler that affects mitochondrial membrane potential causing mitochondrial fragmentation. Images of CCCP-treated mitochondria do not support enhanced fission questioning the effect of treatment.
4. Conclusions need to be better supported by experimental data.
5. The manuscript could significantly benefit from editing throughout.

Specific comments

Figure 1a-c. The legend indicates n=3-6 while quantification does not have any error bars. Please clarify what n represents.

Images of mitochondria in Fig. 1d are of a very poor quality.

Fig. 1f: the complete mitochondrial degradation happens within 5 min after CCCP treatment and stays the same over 2 hrs. This is a very rapid loss of ~80% of all mitochondria in cells that is difficult to understand. It will be helpful if the authors provide images of mitophagy in these cells within this time frame (5 min) using IH, EM and WB to show activation of the corresponding pathways. Error bars in Fig. 1f are missing.

Fluorescence images for experiments described in Fig. 1g should be included. It is not clear whether MTDR could be completely washed out of mitochondria to allow the evaluation of newly synthesized mitochondria.

The authors should consider measuring mitoDNA/nuclear DNA ratio as additional outcome for biogenesis.

Figure 2: JC1 dye is used to determine mitochondrial membrane potential. Active mitochondria exhibit brighter red fluorescence signal compared to mitochondria with lower membrane potential which fluoresce green. Changes in the red/green fluorescence signal ratio can be used to determine healthy versus depolarized mitochondria. Based on experiments presented in Fig. 2, all cells before CCCP treatment demonstrate significant loss of membrane potential (high green signal). It is unclear why this is happening, and how treatment with the uncoupling agent CCCP increases membrane potential. The use of JC1 dye alone can't support the author's conclusions regarding biogenesis.

Figure 3: the quality of WB is poor. It is not clear why patterns of AMPK activation are different in WT vs E50K cells

Figure 4a: based on the representative image, treatment with BX795 in WT cells appears to cause more aggregates contradicting the authors' statement.

A second measure other than Tom 20 is needed to support the authors' conclusions presented in Fig. 4c,d

Figure 5: EM can't be used for mitochondrial mass measurements. Please consider measuring mitoDNA copy number

Figure 6: Neuroprotection should be showed in the presence of a stressor . The authors assessed neuroprotection only in natural "no stress" condition where BX795 treatment yielded increased cell number in both control and glaucomatous hRGCs (with no statistics of cell survival between these two groups). This experimental plan detaches the first half of the work where the effect of CCCP is assessed from the later part where TBK1 is inhibited. The outcomes will be stronger if the TBK1

inhibition induced neuroprotection in the presence of CCCP.

Furthermore, from the Seahorse curves, the increased OCR (claimed in line 202) is not clear. Basal OCR was clearly unchanged in Figure 6(a and b) with application of BX795. Aspect ratio in Figure 5g is a weak indication of intact mitochondrial function. If BX795 indeed increases OCR, it should be presented. If BX795 increases mitochondria number per cytoplasm area but does not increase basal OCR (normalized by cell area, more or less similar to cytoplasm area) in Seahorse, it should decrease mitochondrial oxygen consumption rate. This needs further explanation.

Rigor and reproducibility:

Multiple concerns with data presentation and quantification. For example, standard error or standard deviation and individual data points are not presented in Figure 1(c,d), Figure 3(f-i), and Figure 4(f, h-l). It is also unclear what the following statement "average band intensity quantification w.r.t. actin/GAPDH and DMSO" implies. If the average value of a specific condition's bands' grayscale value was divided by the average grayscale values of actin/GAPDH bands of DMSO treated group for a panel, then the data need to be presented differently. Each band should be normalized by its respective actin/GAPDH band and then the mean with standard deviation/error and individual data points should be presented. In Figure 3(e), AMPK α for at 24hr and pAMPK α at 6hr for H7-hRGCE50K visually does not corroborate the data presented at Figure 5(g,h). This emphasizes the presentation of SD/SE and data points in curves like in Figure 5(g). Better representative western blot images should also be presented.

Response to Reviewers' comments:

Reviewers' comments:

Reviewer #1 (Remarks to the Author):

Dear Authors,

Your manuscript on mitochondrial biogenesis in cultured human RGCs represents a well-written account of a well-conducted study. I have a few comments and suggestions for you to consider, as listed below.

One general comment first. You state in your discussion that AMPK is a 'great' drug target. I would maybe say it represent an 'interesting' potential drug target. Since this seems to be a key take-away message from your paper, please consider if a more elaborate discussion on this would add to your manuscript. I know there are phase II/III studies on the way with direct AMPK activators and that indirect activators (metformin) have been in clinical use for decades and are being repurposed for e.g. cancer. There are some interesting differences to consider in the context of your work since you use a drug that indirectly activates AMPK but do not inhibit mitochondrial respiration/ATP production as I assume metformin does.

We really appreciate your suggestion and critical comments. We have now added text in the discussion to account for the clinical development of AMPK activators. Below text in the manuscript reflect above changes.

Line 362-367: "There are several AMPK activators currently under clinical trials for metabolic diseases¹. Among them metformin is widely used for type 2 diabetes, however the drug functions through inhibiting mitochondrial complex I which reduces ATP production¹. Our study provides an indirect activation mechanism of AMPK by targeting TBK1 without compromising mitochondrial function. Thus, TBK1 provides a very interesting target for developing neuroprotection through restoring energy homeostasis which remains unexplored for CNS disorders."

Comments to the analysis and presentation of the data, figure by figure:

We have now rigorously revised the manuscript where we repeated western blots to present high-quality images with robust statistical analysis. All the data in the revised manuscript are presented with appropriate statistical analysis. Please see below response to each comment.

1. Fig. 1: Is it reasonable to infer that no difference in Tom70/Tom20 expression among the cell lines are indicative of no difference in total mitochondrial mass? Has this been shown experimentally? Please cite relevant studies.

Tom70/Tom20 proteins are mitochondrial protein and provide an indirect measurement of mitochondrial mass. However, we agree with the reviewer that a more direct measurement of mitochondria mass is necessary. In the revised manuscript, we have performed mitochondrial DNA copy number measurements (Fig. 1e, f) in addition to mitochondrial immunofluorescence measurements to validate mitochondrial mass changes under CCCP damage. Below text in the manuscript reflects above changes.

Line 111-115: "We further validated this observation by qPCR-based measurement of mitochondrial DNA copy number relative to nuclear DNA and found WT hRGCs maintained mitochondrial mass similar to the control condition while glaucomatous E50K hRGCs showed moderate reduction at the longer CCCP treatment (Fig. 1e, f). These data suggest hRGCs possess stringent MQC mechanisms for maintaining total mitochondrial mass even with acute damage."

2. Fig. 1: Please indicate the SD for each datapoint presented in panels b and c.

Figure 1b, c was the western blot measurements of the mitochondrial protein Tom70 which we have replaced with the more direct mitochondrial DNA copy number measurements (Fig. 1e, f). However, in the revised manuscript, all the western blots and other quantifications are presented with standard error of mean with appropriate statistical comparisons.

3. Fig. 1, panels b and c: The mass stayed similar, you state. But the level of TOM70 seem to be dynamic and no statistical analysis is offered. Please explain and revise the manuscript to clarify this point.

We have now added statistical analysis to the mitochondrial mass measurements (Fig. 1c-f). Our data shows total mitochondrial mass under CCCP damage fluctuates around the DMSO control condition. In our successive experiments in Fig. 1h, 1m and Fig. 2 we have shown hRGCs efficiently degrade damaged mitochondria and simultaneously activate biogenesis (Fig. 1m; Fig. 2) to maintain a homeostasis under damage. Thus, it is expected to have fluctuating mitochondrial total mass under acute CCCP damage.

4. Fig. 1: At this point in the manuscript, the rationale for choosing 10 μ M CCCP is not clear to me. How was the concentration chosen and what was the goal?

In our prior work we had done extensive dose response study and identified 10 μ M CCCP as the minimum concentration to have maximum mitochondrial degradation in hRGCs². In the literature, 10 μ M CCCP is widely used for inducing mitophagy. Please see below text in the manuscript for justification.

Line 100-104: "Carbonyl cyanide m-chlorophenylhydrazone (CCCP) has been widely used in literature at 5-20 μ M concentrations to induce mitochondrial damage^{3,4}. This has been shown, 10 μ M of CCCP is effective in changing mitochondrial morphology within few seconds in cultured cells⁴. We have previously performed a dose response and

showed 10 μ M CCCP as minimum dose to have maximum mitochondrial degradation in hRGCs².”

5. Fig. 1, panel f: I'm confused here. Using the MTDR dye you say that mitochondrial degradation is increased by CCCP. Is that not the same as mitochondrial mass is decreasing? Please explain and revise the manuscript to clarify this point. Please link your reply to comments 1 and 3 above. Please also explain the choice of statistical analysis. I would tend to think a two-way/repeated measures ANOVA would be more appropriate here since you have two dependent variables.

We thank reviewer for this important question. Please note, Fig. 1f is 1h now. In the revised manuscript total mitochondrial mass at different timepoint of CCCP treatment or DMSO are measured by mitochondrial IF (against Tom20) and mitochondrial DNA copy number by qPCR. These readouts cannot distinguish between the degraded or newly synthesized mitochondria, rather it measures the total mitochondrial mass at that condition. Not seeing a dramatic decrease (Fig. 1c-f) in these experimental designs is an indication of mitochondrial biogenesis to compensate the loss in hRGCs (related to your comments 1 and 3). In Fig. 3, we have identified activation of the mitochondrial biogenesis pathway under CCCP damage to further support this notion.

In Fig. 1h (1f in initial manuscript) we are measuring only the damaged mitochondrial population that are under degradation. To show mitochondrial degradation under CCCP damage, we used an assay previously developed by us². In this experiment, we labeled mitochondria in healthy cells with MTDR dye and then washed out the dye, followed by CCCP treatment and then measured mitochondrial mass in single cells by flow cytometer (Fig. 1g). In this study design, we were able to measure MTDR labelled mitochondrial mass in independent samples after different CCCP treatment time points. This allowed us to track degradation of the labeled population while excluding any newly synthesized mitochondria, as there is no dye in the media for new mitochondria to incorporate.

Please see line 130-137 in the manuscript for above clarification.

For statistical analysis in Fig. 1h, each condition is an independent data set, we have changed the presentation style to bar plot with individual data point to better reflect that. We compared average fluorescence intensity of the MTDR labelled hRGCs between DMSO vehicle control and the individual CCCP timepoints using a student's *t-test* as they are all individual samples, not repeated measurements from one sample. Repeated measure analysis is not suitable as we did not collect cells at different time points from same culture rather, we used independent culture well for each condition and time points.

6. Fig. 1, panel g: Please back up the statement regarding an increase in the MTDR signal to baseline levels with a statistical analysis (two-way/repeated measures ANOVA perhaps?)

In the revised manuscript, Fig. 1g is 1m and now we have added statistical analysis comparing the individual timepoints after CCCP wash to the control condition. In this experimental design, we kept MTDR during CCCP treatment and after CCCP wash, so that the newly synthesized mitochondria can incorporate MTDR which are measured by flow cytometry. We used independent samples for each condition for individual cell type and used Student's *t*-test to compare between two conditions. Our data showed MTDR labelled mitochondrial mass is significantly low right after CCCP treatment compared to DMSO. However, over time mitochondrial mass increased to the DMSO level and showed no significant difference by *t*-test, for H7-E50K hRGCs we saw slight increase compared to the control. Please see Fig. 1m for detail on statistics. Repeated measure analysis is not suitable for the reason mentioned in comment 5.

7. Fig. 1 and other figures with WB data: Please show the entire blots in the supplementary section including the MW markers.

We have now added the full western blot images with MW markers corresponding to each figure in Supplementary figure 6.

8. Fig. 2: As a general note for fig. 2 and other figures, I would like to applaud you for showing the individual data points. This is something that needs to become the standard way of showing data for what I think are obvious reasons.

Thank you very much for your kind comment.

9. Fig. 2: In relation to the two-way ANOVAs that you do, it would be interesting to see the analysis in the contribution of the two dependent variables to the total variation in the dataset. Please consider presenting this analysis.

In the revised manuscript we added some more JC-1 experiments to address other comments. Each condition in this experiment is from an independent biological sample. In the revised manuscript we have compared two conditions for each cell line which is appropriate for Student's *t*-test.

10. Fig. 3: As suggested for the plots in fig. 1, please consider showing the error bars in panels f-l and do a repeated measures ANOVA.

We have added error bars for each of the figures. In the revised manuscript we present bar plots with all the datapoints. All the conditions are from independent cell cultures, not collected from same culture at different time points. Hence a comparison between two independent datasets for each cell type is done using Student's *t*-test.

11. Fig. 4: Again, nicely layed out figure. I will just repeat myself here and ask that you consider a statistical analysis for the data presented in panels f and h-l.

In the revised manuscript, we have added all the datapoints and performed appropriate statistical analysis. Please see statistics section for detail.

12. Fig. 5: For this figure you chose to present the data differently employing what looks like violin plots in panel g. For panels e and f, I assume the vertical line represents the mean, but this is not stated. Please elaborate on the choice of data visualization for this figure.

In the revised manuscript we have presented these data in bar plots, similar to the other figures, with appropriate statistical analysis.

13. Fig. 6 and suppl. Figures, no comments.

We have performed additional Seahorse experiments in Figure 6 to address reviewer 3's comments, please see response to reviewer 3's comment.

Reviewer #2 (Remarks to the Author):

In this manuscript, Surma et al aim to show that human retinal ganglion cells (hRGCs) efficiently generate mitochondria under mitochondrial damage conditions, and to demonstrate that TBK1 inhibition activates mitochondrial biogenesis and exerts neuroprotective effects in both WT and glaucomatous RGC neurons by pharmacological inhibition of TBK1 or activation of AMPK for mitochondrial production and cell survival. Although the current study is potentially interesting, it suffers from significant data inconsistency and lack of in-depth investigation e.g., upon mitochondrial clearance and TBK inhibition.

We thank reviewer for critically reviewing our manuscript and asking several vital questions as addressing them significantly improved our manuscript. We have added an extensive amount of experiments and data analysis to address each of your comments. In the revised manuscript, we have shown the consistent changes in mitochondrial mass by immunofluorescence and mitochondrial DNA copy number measurements under CCCP damage or TBK1 inhibition mediated mitochondrial biogenesis. We have performed additional JC1 labelling experiments and found E50K hRGCs struggle to regain mitochondrial homeostasis after the CCCP damage compared to the WT cells, which addresses the relevance of using glaucomatous E50K hRGCs for this experiment. We have repeated all the western blots and presented quantifications with statistical analysis which reproduced the conclusions from the previous submission. In the revised manuscript multiple lines of experiments showed consistent change in mitochondrial homeostasis, metabolism and cell viability with statistical analysis. Following reviewer's suggestion, we now see BX795 mediated activation of mito-biogenesis can provide neuroprotection to both WT and E50K hRGCs under CCCP damage. To address reviewer's comment if mitophagy is happening in hRGCs within 5min treatment of CCCP, we have shown the increase in LC3B lipidation which is the gold standard for measuring mitophagy activation (please see comment 2 for details).

Specific comments are appended:

1. Figure panels should be better arranged. Currently data quantification is separated from figure data. For example, Fig 1a (top two panels) and 1b should be labelled as Fig 1a; Fig 1a (bottom two panels) and Fig 1c should be labelled as Fig 1b. Such changes should be made throughout the manuscript.

We have now replaced the western blots (Fig. 1a-c) with immunofluorescence and mtDNA copy number measurements to quantify mitochondrial mass. We have presented images and corresponding data quantification in sperate panel to clearly explain those data in the result section. For example, in figure 3e we have western blot images for AMPK α , p-AMPK α , PGC1 α , p-PGC1 α and actin loading control and in separate panels we have quantification for the phoshpo-protein to total protein ratio for each RGC type. Having all the blots for each cell type and corresponding protein quantifications in one panel will be difficult to explain changes corresponding to each protein.

2. In Fig 2, how does mitochondrial clearance occur in the hRGCs under CCCP treatment? It is necessary to provide direct evident to support this. Mitophagy should be tested here.

CCCP is a robust mitochondrial uncoupler and widely used in the literature for inducing mitophagy. We have now performed LC3B lipidation experiment and observed activation of mitophagy withing 5min of CCCP treatment. Please see below text from the manuscript for additional explanation

Line 137-145: "Damaged mitochondria are degraded by lysosomes via LC3B autophagic flux, a process known as mitophagy⁵. The hallmark for activation of the mitophagy pathway is an increase in lipidated LC3B. Mitophagy complex formation on mitochondria depends on the LC3B-I (non-lipidated, 16 kDa) to LC3B-II conversion (lipidated, 14 kDa), which migrates faster during gel electrophoresis and enrichment of the second band in western blot is a classic measurement of the induction of mitophagy⁶. We hypothesized that induction of mitochondrial damage would lead to mitophagy by increasing LC3B lipidation. Indeed, we observed rapid induction of mitophagy within 5 min of CCCP treatment as shown by increased lipidated LC3B form in both WT (Fig. 1i, j) and E50K hRGCs (Fig. 1k, l)".

In a separate project, we are performing in-depth analysis to resolve the mitophagy mechanisms in hRGCs and potential defects for E50K glaucomatous mutation. This study includes the identification of specific mitophagy players under CCCP damage, such as which mitophagy adaptor (P62, NDP52, OPTN, NBR1) or receptor (BNIP3L, FUNDC1, GABARAPs, AMBRA1) proteins are involved and mechanisms of mitophagy in hRGCs. This study will be published soon elsewhere; however, it is beyond the scope of current manuscript.

3. It seems that mitochondrial (JC1 red) levels in E50K cells are not reduced, compared with WT cells (Fig 2b-d). This raises a question whether E50K mutation is relevant to the following experiments.

We thank reviewer for the critical question. We have performed additional experiments to resolve the question and further supported our conclusions with additional seahorse experiments. Mitochondria in E50K hRGCs are functionally active to meet cellular metabolic need, so it is not surprising to see similar membrane potential to WT cells. However, E50K hRGCs showed reduced capacity to regain basal level polarized to depolarized mitochondrial distribution after an acute damage by CCCP (new experiments, please see below for details).

For JC1 experiment, our goal was to identify if both the wild type and E50K hRGCs produce new mitochondria under the CCCP damage. Thus, in our assay design we maintained JC1 dye in the media before, during CCCP treatment and during imaging so that any newly synthesized mitochondria could be detected de-novo as they incorporate the dye. This has been reflected by the appearance of mitochondria with JC1 red (J aggregates) fluorescence and clearance of acutely damaged JC1 green (J monomer) mitochondria. In the revised manuscript, we presented the intensity ratio of red:green which shows both the WT and E50K hRGCs instantaneously produce mitochondria to compensate for the CCCP induced loss. We have now removed the JC1 green and JC1 red alone quantifications as it might not be appropriate given our data shows E50K hRGCs contain lower mitochondrial mass compared to the WT cells (Fig. 4a, b; Fig. 5b; Supp Fig. 3).

To address if E50K mutation defective restoring mitochondrial homeostasis, we performed JC1 labelling after CCCP wash and rescue. The experimental design is explained in Fig. 2d. Indeed, in this experiment we saw E50K hRGCs possess less red:green mitochondria after CCCP wash compared to the DMSO control, but WT hRGCs have negligible effect (Fig. 2e, f). However, under control condition E50K hRGCs possess similar red:green (healthy:damaged) mitochondrial population compared to the WT cells. This suggests at the basal level both the WT and E50K hRGCs maintain similar mitochondrial membrane potential, however, after an acute stress E50K hRGCs struggle to restore the homeostasis. This makes sense as membrane depolarization would lead to less efficient ATP production, but our new seahorse analysis revealed E50K hRGCs consume more oxygen (OCR) with a higher mito-ATP production rate (Fig. 6a-e) to meet the metabolic demand. Hence, mitochondria membrane polarity shouldn't be reduced, rather E50K hRGCs suffer more ATP production load per mitochondrion as it has less mitochondrial mass.

4. It is not clear how the treatment was done in Fig 4b, and the Fig legend does not correspond to the figure.

In the revised manuscript this figure is Supplemental figure 5 and we have added treatment condition in detail in the legend. In the previous figure legend, the label was after the description. We have clarified and added description after the label throughout.

5. Fig 4a-b, why could OPTN aggregates be dissolved in the E50K-hRGCs with TBK1 inhibition?

The detail mechanism is yet to be discovered, however change in electrostatic interactions are likely be the reason for OPTN-E50K aggregate dissolution. TBK1 interacts with OPTN and phosphorylates multiple residues that regulate its function⁷. The interaction between OPTN-TBK1 is maintained by a balance of positive and negative electrostatic forces that helps the association and dissociation of the complex⁸. In this complex structure, OPTN E50 residue interacts with the positively charged TBK1 K694 residue and is neutralized by the two negatively charged E46 of OPTN and E698 residue of TBK1. With the change in glutamic acid to lysine at the 50th position of OPTN (E50K), the positively charged OPTN K50 residue in the mutant protein has a strong interaction with the negatively charged TBK1 E698 residue. This results in around 40-fold higher binding affinity of OPTN-E50K to TBK1 compared to wild-type OPTN. This increased affinity, results in the failure of dissociation of OPTN-TBK1 complex, leading to aggregate formation⁸. TBK1 inhibition can perturb the interaction with OPTN owing to the lack of OPTN phosphorylation leading to reduced electrostatic interaction between the negatively charged phosphorylated OPTN residues and positively charged TBK1 residues. Thus, TBK1 inhibition can counter the enhanced OPTN(E50K)-TBK1 interaction leading to aggregate dissolution. However, the detailed mechanisms of how TBK1 inhibition dissolves OPTN-E50K aggregates are yet to be discovered.

It has also been shown biochemically that HEK293 cells expressing OPTN(E50K) protein formed more insoluble fraction when purified, which got dissolved under TBK1 inhibition by BX795⁹. In the revised manuscript we have moved this aggregate dissolution figure to Supplemental figure 5 as this phenomenon could be independent of BX mediated activation of mitochondrial biogenesis which is the focus of figure 4.

6. WT and E50K cells appear respond to TBK1 inhibition differently in Fig 4c-d, but in Fig 4e-f, the two types of cells behave similarly. How could this be the case?

In Fig. 4c-d we measured the long term (24h, 48h, 72h) effect of TBK1 inhibition by BX treatment followed by mitochondrial immunofluorescence (IF). In Fig. 4e-f we examined the early response (1h, 6h, 18h, 24h) for TBK1 inhibition on mitochondrial mass by western blot (WB) against mitochondrial Tom70 protein. During the early response (1h, 6h) we saw mitochondrial mass initially increased, while longer treatment showed minimal change presumably due to the establishment of homeostasis. The difference could also be due to the difference between IF and WB readouts as using WB to measure mitochondrial mass could be complicated by changes in protein homeostasis, hence is an indirect measurement.

To resolve above issue, in the revised manuscript we have performed mitochondrial IF and mitochondrial DNA (mtDNA) copy number measurements by qPCR at the similar BX treatment time points (1h, 6h, 18h, 24h). Both the measurements showed consistent increase in mitochondrial mass for the WT and E50K hRGCs upon BX treatments (Fig. 4a-d). We have replaced WB with the mtDNA copy number measurements as it is a more direct readout of mitochondrial mass.

7. The immunoblot data in Fig 4e, g does not appear convincing. The increase in Tom70 levels seems transient, and the enhanced levels in PGC-1a, b or pAMPK/AMPK are modest or transient.

In the revised manuscript, as mentioned above we have replaced Tom70 western blots with more direct mtDNA copy number measurements. We have performed additional experiments to measure AMPK α and PGC1 α activation by western blots and presented all the data with error bars and appropriate statistical analysis.

8. In Fig 5, BX795 treatment seems to give rise to significant effects on mitochondrial levels in E50K cells, but not in WT cells. This is somewhat confusing, and inconsistent with the data in Fig 4 and Fig 6.

We thank the reviewer for pointing out this difference. After analyzing the data for its distribution, we found the individual data sets for Fig. 5 are non-normal. Hence, in the revised manuscript we used Mann Whitney U test to compare between the two independent data sets. This analysis revealed consistent increase in the mitochondrial coverage or mass (mito area/cell area) for WT (p-value: 0.08) and E50K hRGCs (p-value <0.0001). Additional seahorse experiments in figure 6 further support this observation as increased mitochondrial mass in the E50K hRGCs lowers mitoOCR (Fig. 6e) to maintain energy homeostasis as per mitochondrion needs to produce less ATP.

9. In Fig 6, BX795 treatment appears to promote the survival for both WT and E50K hRGCs, but not for the iPSC cells, however it reduces caspase activity in iPSC cells.

Apoptotic RGC death is the hallmark for glaucoma irrespective of the causal factors¹⁰. So, to better estimate the pro-survival effect of BX795, in the revised manuscript we measured apoptosis in the basal as well as under CCCP damage. Indeed, we observed consistent reduction of apoptosis in the WT and E50K hRGCs after 24h BX795 treatment (Fig. 6i, k). Remarkably CCCP induced apoptosis in the E50K hRGCs after 48h treatment which was mitigated by the BX795 (Fig. 6l) showing its neuroprotection effect.

**Minor points:
Page numbers should be added.**

This has been done

What does w.r.t. stand?

W.r.t stands for with-respect-to, we have minimized the use of this abbreviation, full name is mentioned in line 541

Reviewer #3 (Remarks to the Author):

General comments

1. The authors have to provide the justification for the use of 10uM of CCCP for treatment. It is unclear why and how these conditions were selected.

In our prior work we had done extensive dose response study and identified 10 μ M CCCP as the minimum concentration to have maximum mitochondrial degradation in hRGCs². In the literature 10 μ M CCCP is also widely used for inducing mitophagy. Please see below text in the manuscript for justification.

Line 100-104: "Carbonyl cyanide m-chlorophenylhydrazone (CCCP) has been widely used in literature at 5-20 μ M concentrations to induce mitochondrial damage^{3,4}. This has been shown, 10 μ M of CCCP is effective in changing mitochondrial morphology within few seconds in cultured cells⁴. We have previously performed a dose response and showed 10 μ M CCCP as minimum dose to have maximum mitochondrial degradation in hRGCs²."

2. The quality of mitochondrial staining using fluorescence dyes is unacceptable. Images throughout the manuscript are exceptionally poor not allowing to visualize individual mitochondria.

We have reproduced and replaced prior images with improved mitochondrial immunofluorescence (IF) imaging throughout the manuscript (Fig. 1a, 1b, 4a, Supp Fig 3 and Supp Fig 5a). Please note, unlike some large adherent cells, RGCs possess a small cytoplasmic area around the nucleus and a highly convoluted mitochondrial network whose structure is difficult to resolve by confocal IF. By mitochondrial IF our goal is to measure the mitochondrial mass per cell, but to resolve the structural aspects of mitochondria we have performed in-depth electron microscopy analysis (Fig. 5).

3. CCCP is an uncoupler that affects mitochondrial membrane potential causing mitochondrial fragmentation. Images of CCCP-treated mitochondria do not support enhanced fission questioning the effect of treatment.

In the revised manuscript, in Fig. 1a WT hRGCs show more thread like mitochondrial network, however in presence of 10 μ M CCCP this appearance becomes more punctate as an indication for mitochondrial fragmentation. Similar changes were also observed for E50K hRGCs (Fig. 1b), however E50K mitochondria appeared to be slightly punctate

even in the DMSO control condition. Bright red mitochondrial puncta are also evident in the Figure 2b where hRGCs are treated with 10 μ M CCCP and imaged in presence of JC1 dye. Apart from mitochondrial morphology, acute degradation of MTDR labeled mitochondria (Fig. 1h), degradation of JC1 labelled damaged (green) mitochondria and activation of the biogenesis pathway genes (Fig. 3) in presence of 10 μ M CCCP provide robust evidence of the effectiveness of this treatment.

4. Conclusions need to be better supported by experimental data.

We have added extensive amount of new experiments to consolidate our conclusions (new experiments: Fig. 1e, f; 1i-l; 2d-f; 4a-d; 5c-e; 6a-e; 6i-l; Supp Fig. 3; Supp Fig. 4; Supp Fig. 6). All the data are quantified and presented with error bars and statistical analysis. We have reproduced all the western blots and presented high quality images with statistical analysis and error bars. We thank reviewer for critically reviewing our manuscript as this rigorous revision reproduced and strengthened major conclusions with mechanistic detail.

5. The manuscript could significantly benefit from editing throughout.

We have thoroughly edited the manuscript.

Specific comments

Figure 1a-c. The legend indicates n=3-6 while quantification does not have any error bars. Please clarify what n represents.

In the revised manuscript we have replaced mitochondrial Tom70 western blots (1a-c in previous submission) with the mtDNA/nDNA qPCR measurements which is a more direct measurement for mitochondrial mass. This also addresses your next comment. Now we have presented all the western blots and other quantifications with error bars and statistical analysis. Each n is one independent sample or cell for imaging. So, for example, in figure 3f-i, n=3 corresponds to three independent biological repeats for each timepoint.

Images of mitochondria in Fig. 1d are of a very poor quality.

We have replaced those images with better quality images which are now in Fig. 1a, b. Also, please see response to your general comment 2.

Fig. 1f: the complete mitochondrial degradation happens within 5 min after CCCP treatment and stays the same over 2 hrs. This is a very rapid loss of ~80% of all mitochondria in cells that is difficult to understand. It will be helpful if the authors provide images of mitophagy in these cells within this time frame (5 min) using IH, EM and WB to show activation of the corresponding pathways. Error bars in Fig. 1f are missing.

In the revised manuscript Fig. 1f is 1h. In Fig. 1h we are measuring only the damaged mitochondrial population that are under degradation. To show mitochondrial degradation under CCCP damage, we used an assay previously developed by us² where we labeled mitochondria in healthy cells with MTDR dye and then washed out the dye followed by CCCP treatment and measured mitochondrial mass in single cells by flow cytometer (Fig. 1g). In this study design, we are able to measure MTDR labelled mitochondrial mass in independent samples at different CCCP treatment time points. This allowed us to track the labeled population degradation, excluding any newly synthesized mitochondria as there is no dye in the media for new mitochondria to incorporate. So even though we see around 50-80% reduction in mitochondrial mass from 5min-2h after CCCP treatment that is only the pre-labelled population, not the total mitochondrial mass. The data from Fig. 1m and 2b, c suggests hRGCs activate biogenesis instantaneously to compensate the mitochondrial loss.

For mitophagy, we have now performed LC3B lipidation experiment and observed activation of mitophagy withing 5min of CCCP treatment. Please see below text from the manuscript for additional explanation.

Line 137-145: "Damaged mitochondria are degraded by lysosomes via LC3B autophagic flux, a process known as mitophagy⁵. The hallmark for activation of the mitophagy pathway is an increase in lipidated LC3B. Mitophagy complex formation on mitochondria depends on the LC3B-I (non-lipidated, 16 kDa) to LC3B-II conversion (lipidated, 14 kDa), which migrates faster during gel electrophoresis and enrichment of the second band in western blot is a classic measurement of the induction of mitophagy⁶. We hypothesized that induction of mitochondrial damage would lead to mitophagy by increasing LC3B lipidation. Indeed, we observed rapid induction of mitophagy within 5 min of CCCP treatment as shown by increased lipidated LC3B form in both WT (Fig. 1i, j) and E50K hRGCs (Fig. 1k, l)."

In a separate project, we are performing in-depth analysis to resolve the mitophagy mechanisms in hRGCs and potential defects for E50K glaucomatous mutation. This study includes the identification of specific mitophagy players under CCCP damage, such as which mitophagy adaptor (P62, NDP52, OPTN, NBR1) or receptor (BNIP3L, FUNDC1, GABARAPs, AMBRA1) proteins are involved and mechanisms of mitophagy in hRGCs. This study will be published soon elsewhere; however, it is beyond the scope of current manuscript.

Figure 1f is 1h in the revised manuscript which is presented by bar plots with all the data points and error bars. In the previous submission also, there were error bars on Fig. 1f, they are simply very close to the data points but if you zoom in could see error bars.

Fluorescence images for experiments described in Fig. 1g should be included. It is not clear whether MTDR could be completely washed out of mitochondria to allow the evaluation of newly synthesized mitochondria.

The imaging-based detection of newly synthesized mitochondria by JC1 dye in the presence of CCCP (Fig. 2a-c) is intended to address above comment. Since mitochondrial degradation and biogenesis are happening simultaneously under mitochondrial stress, just observing mitochondria will not be able to detect individual populations. Hence, we had to design experiments such that we can monitor either the degrading or the newly synthesized mitochondrial population. Please note in Fig. 1m (1g in previous) the study design is to have MTDR after the CCCP wash so that the newly synthesized mitochondria incorporate the dye and gradually show higher intensity over time. This readout is not dependent on the MTDR washing out from existing mitochondria. So, for example if newly synthesized mitochondria take-up dye from the existing mitochondria, per cell there will be no increase in MTDR fluorescence intensity.

The authors should consider measuring mitoDNA/nuclear DNA ratio as additional outcome for biogenesis.

We have incorporated this experiment in the revised manuscript (Fig. 1e, f; Fig. 4c, d).

Figure 2: JC1 dye is used to determine mitochondrial membrane potential. Active mitochondria exhibit brighter red fluorescence signal compared to mitochondria with lower membrane potential which fluoresce green. Changes in the red/green fluorescence signal ratio can be used to determine healthy versus depolarized mitochondria. Based on experiments presented in Fig. 2, all cells before CCCP treatment demonstrate significant loss of membrane potential (high green signal). It is unclear why this is happening, and how treatment with the uncoupling agent CCCP increases membrane potential.

The use of JC1 dye alone can't support the author's conclusions regarding biogenesis.

We thank the reviewer for asking this critical question. In the revised manuscript we have added additional experiments to resolve the mitochondrial biogenesis aspect. Please note, it is widely observed in cultured healthy brain neurons that mitochondria possess significant amount of depolarized (JC1 green) and polarized (JC1 red) membranes¹¹. So, a similar observation in the cultured hRGCs is not surprising. In Figure 2a-c, we had acquired images in presence of JC1 before CCCP addition and then in presence of CCCP to detect changes in green and red mitochondria population. In case of a poor homeostasis such as slow degradation and biogenesis pathways, one would expect to see more JC1 labelled green and less red mitochondria under CCCP as you suggested. However, for an efficient homeostatic system, such as for hRGCs, it is expected that cells will quickly remove the damaged mitochondria (green) and produce healthy mitochondria (red) which we also have observed in Fig. 2a-c. In the revised manuscript we presented the intensity ratio of red:green which shows both the WT and E50K hRGCs instantaneously produce mitochondria to compensate for the CCCP induced loss. We have removed the JC1 green and JC1 red alone quantifications as it might not be appropriate given our data shows E50K hRGCs contain lower mitochondrial mass compared to the WT cells (Fig. 4a, b; Fig. 5b; Supp Fig. 3).

To further investigate the mito-biogenesis in hRGCs, in the revised manuscript we have performed additional experiments where we first treated cells with CCCP then washed and allowed cells to recover in presence of JC1 media only (Fig. 2d). Our rationale here is during the recovery step hRGCs will produce healthy mitochondria with spontaneous distribution of polarized (red) and depolarized (green) mitochondrial membrane as observed before treatment. Indeed, we observed WT hRGCs regained the homeostasis (Fig. 2f) further supporting mitochondrial biogenesis after CCCP stress.

In addition to above, we have several lines of evidence beyond JC1 dye that hRGCs activate biogenesis under damage such as, direct measurement of newly synthesized mitochondria by flow cytometry (Fig. 1m), and activation of the biogenesis pathway proteins (Fig. 3).

Figure 3: the quality of WB is poor. It is not clear why patterns of AMPK activation are different in WT vs E50K cells

We have repeated these western blots and presented high quality images with quantifications and statistical analysis. Our data shows activation of AMPK α for both the WT and E50K hRGCs under CCCP treatments.

Figure 4a: based on the representative image, treatment with BX795 in WT cells appears to cause more aggregates contradicting the authors' statement.

Immunofluorescence of OPTN shows granular appearance even in the WT hRGCs. *OPTN*^{E50K} protein, however, shows bigger puncta, which get dissolved under BX795 mediated TBK1 inhibition. In the revised manuscript, we have presented images and quantifications that better reflect the above observations. Please note we have moved this part of the figure to Supp Figure 5 as the observation is not central to the mitochondrial biogenesis pathway which is the focus of figure 4.

A second measure other than Tom 20 is needed to support the authors' conclusions presented in Fig. 4c,d

We have now incorporated mtDNA/nDNA qPCR measurements to support changes in mitochondrial mass under BX795 treatments (Fig. 4c, d).

Figure 5: EM can't be used for mitochondrial mass measurements. Please consider measuring mitoDNA copy number

Please see above as we have incorporated mitoDNA copy number measurements. We have used EM images to resolve the mitochondrial structural differences between the WT and E50K hRGCs and structural improvement under BX795. In the revised manuscript we have added measurements for mitochondrial perimeter, major axis, minor axis (Fig. 5c-e) and identified E50K hRGCs possess more swelled mitochondria and the swelling has been mitigated by BX795 drug.

Figure 6: Neuroprotection should be showed in the presence of a stressor. The authors assessed neuroprotection only in natural “no stress” condition where BX795 treatment yielded increased cell number in both control and glaucomatous hRGCs (with no statistics of cell survival between these two groups). This experimental plan detaches the first half of the work where the effect of CCCP is assessed from the later part where TBK1 is inhibited. The outcomes will be stronger if the TBK1 inhibition induced neuroprotection in the presence of CCCP.

We thank reviewer for this very important suggestion. In the revised manuscript we have measured cellular apoptosis under CCCP and if that is mitigated by BX795. Apoptotic RGC death is the hallmark for glaucoma irrespective of the causal factors¹⁰. So, to better estimate the pro-survival effect of BX795, we measured apoptosis in the basal as well as under CCCP damage. Indeed, we observed consistent reduction of apoptosis in the WT and E50K hRGCs after 24h BX795 treatment. Remarkably CCCP increased apoptosis in the E50K hRGCs after 48h treatment which was mitigated by the BX795 showing its neuroprotection effect. We have previously published that hRGCs don't show toxic effects in the presence of CCCP² presumably due to their highly efficient mitochondrial homeostasis mechanisms as shown here. Please see below text in the manuscript for justifications.

Line 320-330: “Apoptotic RGC death is the hallmark for glaucoma irrespective of the causal factors¹⁰. To test if BX795 promotes a pro-survival effect, we measured apoptosis master regulator Caspase-3/7 activity in the WT and E50K hRGCs under basal and CCCP induced damage. Remarkably, we observed 24h of BX795 treatment reduced Caspase activity for the WT and E50K hRGCs both under basal and CCCP damage (Fig. 6i, k). We did not observe any increase in Caspase activity for the WT hRGCs even after 48h of CCCP treatment with no additional effect of BX795 (Fig. 6j) presumably due to the existence of stringent MQC mechanisms. However, glaucomatous E50K hRGCs showed strong increase in caspase activity post 48h of CCCP treatment which is mitigated by the BX795 treatment (Fig. 6l) as evidence for neuroprotection. Furthermore, 48h of BX795 also reduced basal Caspase activity in E50K hRGCs as an indication for sustained neuroprotection under glaucomatous condition.”

Furthermore, from the Seahorse curves, the increased OCR (claimed in line 202) is not clear. Basal OCR was clearly unchanged in Figure 6(a and b) with application of BX795. Aspect ratio in Figure 5g is a weak indication of intact mitochondrial function. If BX795 indeed increases OCR, it should be presented. If BX975 increases mitochondria number per cytoplasm area but does not increase basal OCR (normalized by cell area, more or less similar to cytoplasm area) in Seahorse, it should decrease mitochondrial oxygen consumption rate. This needs further explanation.

We sincerely thank the reviewer for asking critical questions to link the mitochondrial structure and functional changes. To have additional evidence that BX795 does not alter basal OCR, we performed the ATP rate and glycolytic rate assays and observed similar OCR between the DMSO and BX795 treatments (Fig. 6a, c) supporting similar OCR at the basal level in mito-stress test results (Fig. 6f, g). BX795 increased mitochondrial mass in the WT and E50K hRGCs but it did not alter OCR compared to the DMSO control. This observation could be supported if the consumed oxygen (OCR) is distributed among increased number of mitochondria which will lead to reduced mitoOCR under BX795 treatment to maintain energy homeostasis. Remarkably, we have observed reduced mitoOCR/GlycoPER for E50K hRGCs in presence of BX795 in glycolytic rate assay (Fig. 6e) as the reviewer predicted. From ATP rate assay we found E50K hRGCs produce more ATP at the basal level (Fig. 6b) indicating more ATP production load per mitochondrion to meet the metabolic need. Studies have shown that a mild increase of the matrix volume can help expand the inner mitochondrial membrane (IMM), which in turn stimulates the electron transport chain activity that helps increase production of ATP¹²⁻¹⁴. Thus, the mitochondrial swelling in E50K hRGCs is potentially helping them to meet the increased ATP demand on their lesser number of mitochondria. Indeed, by electron microscopy we observed mild mitochondrial swelling in the E50K hRGCs, which is mitigated by BX795 as it increased mitochondrial biogenesis and reduced the ATP production load per mitochondrion.

Please see line 280-318 in the manuscript for above result and explanation.

Rigor and reproducibility:

Multiple concerns with data presentation and quantification. For example, standard error or standard deviation and individual data points are not presented in Figure 1(c,d), Figure 3(f-i), and Figure 4(f, h-l).

We have replaced Fig. 1c, d with mitochondrial copy number measurements with error bar and statistics which are Fig. 1e, f now. The other mentioned plots are for western blot quantifications. In the revised manuscript, we have repeated all the western blots and presented quantifications with error bars and statistics.

It is also unclear what the following statement “average band intensity quantification w.r.t. actin/GAPDH and DMSO” implies. If the average value of a specific condition’s bands’ grayscale value was divided by the average grayscale values of actin/GAPDH bands of DMSO treated group for a panel, then the data need to be presented differently. Each band should be normalized by its respective actin/GAPDH band and then the mean with standard deviation/error and individual data points should be presented.

The “average band intensity quantification” referred to the average values of raw integrated density of protein bands. In the revised manuscript we have clarified quantification method. For each protein of interest (AMPK, PGC1a... etc.) a loading control (Actin or GAPDH) was run using the same membrane. As the reviewer

mentioned, we measured raw integrated density of the bands, and normalized to the corresponding loading control (actin or GAPDH). As the western blots were run on different days, potentially having different signal to noise on one day compared to the others, we further normalized the quantifications (protein/loading control) for each time points to the DMSO (protein/loading control) values for that specific day. We used those final values to calculate the mean and SEM.

Please see line 498-501 in the manuscript for above clarification.

In Figure 3(e), AMPK α for at 24hr and pAMPK α at 6hr for H7-hRGCE50K visually does not corroborate the data presented at Figure 5(g,h). This emphasizes the presentation of SD/SE and data points in curves like in Figure 5(g). Better representative western blot images should also be presented.

As mentioned above, we have repeated all the western blots and presented high quality images and plots with data points and statistics. In the revised manuscript, Figure 3e blot images and corresponding quantifications are in agreement.

References:

1. Steinberg, G. R. & Carling, D. AMP-activated protein kinase: the current landscape for drug development. *Nature Reviews Drug Discovery* vol. 18 Preprint at <https://doi.org/10.1038/s41573-019-0019-2> (2019).
2. Das, A., Bell, C. M., Berlinicke, C. A., Marsh-Armstrong, N. & Zack, D. J. Programmed switch in the mitochondrial degradation pathways during human retinal ganglion cell differentiation from stem cells is critical for RGC survival. *Redox Biol* (2020) doi:10.1016/j.redox.2020.101465.
3. Padman, B. S., Bach, M., Lucarelli, G., Prescott, M. & Ramm, G. The protonophore CCCP interferes with lysosomal degradation of autophagic cargo in yeast and mammalian cells. *Autophagy* **9**, (2013).
4. Miyazono, Y. *et al.* Uncoupled mitochondria quickly shorten along their long axis to form indented spheroids, instead of rings, in a fission-independent manner. *Sci Rep* **8**, (2018).
5. Villa, E., Marchetti, S. & Ricci, J. E. No Parkin Zone: Mitophagy without Parkin. *Trends in Cell Biology* vol. 28 882–895 Preprint at <https://doi.org/10.1016/j.tcb.2018.07.004> (2018).
6. Fujita, N. *et al.* An Atg4B mutant hampers the lipidation of LC3 paralogues and causes defects in autophagosome closure. *Mol Biol Cell* **19**, 4651–4659 (2008).
7. Richter, B. *et al.* Phosphorylation of OPTN by TBK1 enhances its binding to Ub chains and promotes selective autophagy of damaged mitochondria. *Proc Natl Acad Sci U S A* **113**, 4039–4044 (2016).
8. Li, F. *et al.* Structural insights into the interaction and disease mechanism of neurodegenerative disease-associated optineurin and TBK1 proteins. *Nat Commun* **7**, (2016).

9. Minegishi, Y. *et al.* Enhanced optineurin E50k-TBK1 interaction evokes protein insolubility and initiates familial primary open-angle glaucoma. *Hum Mol Genet* **22**, 3559–3567 (2013).
10. Qu, J., Wang, D. & Grosskreutz, C. L. Mechanisms of retinal ganglion cell injury and defense in glaucoma. *Exp Eye Res* **91**, (2010).
11. Buckman, J. F. & Reynolds, I. J. Spontaneous changes in mitochondrial membrane potential in cultured neurons. *Journal of Neuroscience* **21**, (2001).
12. Halestrap, A. P. The regulation of the matrix volume of mammalian mitochondria in vivo and in vitro and its role in the control of mitochondrial metabolism. *Biochimica et Biophysica Acta (BBA) - Bioenergetics* **973**, 355–382 (1989).
13. Armston, A. E., Halestrap, A. P. & Scott, R. D. The nature of the changes in liver mitochondrial function induced by glucagon treatment of rats. The effects of intramitochondrial volume, aging and benzyl alcohol. *Biochimica et Biophysica Acta (BBA) - Bioenergetics* **681**, 429–439 (1982).
14. Okayasu, T., Curtis, M. T. & Farber, J. L. Structural alterations of the inner mitochondrial membrane in ischemic liver cell injury. *Arch Biochem Biophys* **236**, 638–645 (1985).

REVIEWERS' COMMENTS:

Reviewer #1 (Remarks to the Author):

Dear Authors,

Thank you for your extensive revision of the manuscript according to my concerns and suggestions. Well done. I think this is a solid contribution to the field.

Reviewer #3 (Remarks to the Author):

The manuscript was greatly improved. My comments were well addressed. This work now could be accepted for a publication. Congratulations to the team!

Reviewer #4 (Remarks to the Author):

The authors have improved the manuscript.